# Biochar alters hydraulic conductivity and impacts nutrient leaching in two agricultural soils

Danielle L. Gelardi[1,2], Irfan H. Ainuddin[3], Devin A. Rippner[4], Janis E. Patiño[5], Majdi Abou Najm[1], Sanjai J. Parikh[1]

[1]Land, Air and Water Resources, University of California, Davis, 1 Shields Ave, Davis CA 95616, USA

[2]Natural Resources Assessment Section, Washington State Department of Agriculture, Olympia WA 98504, USA

[3]California State University Chico, 400 West First Street, Chico CA 95929, USA

[4]United States Department of Agriculture, Horticulture Crops Research Unit, Prosser WA 99350, USA

[5]Department of Civil and Environmental Engineering, University of California, Davis, 1 Shields Ave, Davis CA 95616, USA

*Correspondence to*: Danielle L. Gelardi (dlgelardi@ucdavis.edu)

**Abstract.** Biochar is purported to provide agricultural benefits when added to the soil, through changes in saturated hydraulic conductivity ($K_{sat}$) and increased nutrient retention through chemical or physical means. Despite increased interest and investigation, there remains uncertainty regarding the ability of biochar to deliver these agronomic benefits due to differences in biochar feedstock, production method, production temperature and soil texture. In this project, a suite of experiments was carried out using biochars of diverse feedstocks and production temperatures, in order to determine the biochar parameters which may optimize agricultural benefits. Sorption experiments were performed with seven distinct biochars to determine sorption efficiencies for ammonium and nitrate. Only one biochar effectively retained nitrate, while all biochars bound ammonium. The three biochars with the highest binding capacities (produced from almond shell at 500 and 800 °C (AS500 and AS800) and softwood at 500 °C (SW500)) were chosen for column experiments. Biochars were amended to a sandy loam and a silt loam at 0 and 2% (w/w) and $K_{sat}$ was measured. Biochars reduced $K_{sat}$ in both soils by 64-80%, with the exception of AS800, which increased $K_{sat}$ by 98% in the silt loam. Breakthrough curves for nitrate and ammonium, as well as leachate nutrient concentration, were also measured in the sandy loam columns. All biochars significantly decreased the quantity of ammonium in the leachate, by 22 to 78%, and slowed its movement through the soil profile. Ammonium retention was linked to high cation exchange capacity and a high oxygen to carbon ratio, indicating that the primary control of ammonium retention in biochar-amended soils is the chemical affinity between biochar surfaces and ammonium. Biochars had little to no effect on the timing of nitrate release, and only SW500 decreased total quantity, by 27 to 36%. The ability of biochar to retain nitrate may be linked to high micropore specific surface area, suggesting a physical entrapment rather than a chemical binding.

Together, this work sheds new light on the combined chemical and physical means by which biochar may alter soils to impact nutrient leaching and hydraulic conductivity for agricultural production.

## 1 Introduction

The ability of biochar to chemically and physically alter soil environments for specific agronomic benefits is the subject of increased investigation, as evidenced by the recent rise in published biochar studies (Web of Science, 2021) and United States

trademark and patent applications listing the word "biochar" (US Patent and Trademark Office, 2021). Biochar, or the carbonaceous material created from the thermochemical conversion of biomass in an oxygen-limited environment (International Biochar Initiative (IBI), 2015), possesses unique chemical and physical properties, determined by variables such as its feedstock, production method, and production temperature. Biochar properties typically include a low bulk density, high porosity, high surface area, reactive surface functional groups, and recalcitrant carbon (Downie et al., 2009). These attributes

make it a promising material for amendment to agricultural soils, as biochar may help improve soil water holding capacity, hydraulic conductivity, and nutrient retention. Despite increased interest and investigation, there remains uncertainty regarding the ability of biochar to deliver these agronomic benefits. While many studies show promising results where nutrient retention and soil water dynamics are concerned (Blanco-Canqui, 2017; Glaser et al., 2002, 2015; Glaser and Lehr, 2019; Haider et al., 2020; Hestrin et al., 2019), others have demonstrated no or only minor effects (Griffin et al., 2017; Jones et al., 2012; Martos

et al., 2020). Several authors have concluded that, due to differences in biochar production parameters and those of the soil environment, material and site-specific investigation is required before conclusions can be drawn about the potential of biochar to provide agricultural benefits (Hassan et al., 2020; Jeffery et al., 2011; Zhang et al., 2016).

The ability of biochar to remove nitrate ($NO_3^-$) and ammonium ($NH_4^+$) from aqueous environments has been widely

investigated, as it may indicate whether biochar can improve crop nutrient use efficiency and supress fertilizer pollution through leaching and volatilization (Clough and Condron, 2010; Peiris et al., 2018). To this effect, batch sorption experiments are commonly carried out to determine the electrostatic affinity between biochars and $NO_3^-$ and $NH_4^+$. The net charge of biochars vary based on their surface functional groups and the degree of protonation, as a function of soil pH and their point of zero charge (PZC). While biochar PZCs between 7 and 10 have been observed (Lu et al., 2013; Uchimiya et al., 2011), the

high number of oxygen-containing functional groups (primarily carboxyl) typically lead to PZCs less than 5 (Peiris et al., 2019; Uchimiya et al., 2011; Wang et al., 2020). As such, the deprotonation of biochar surface functional groups occurs within most agronomic soils (pH ~5-7.5), leading to a net negative charge. It is expected, then, that most biochars would not bind to $NO_3^-$, which exists in the anionic form in aqueous environments, while more readily binding to positively charged $NH_4^+$ ions. Electrostatic repulsion between $NO_3^-$ and biochar has indeed been regularly cited as the reason behind little to no $NO_3^-$ removal

in batch sorption experiments. Zhou et al. (2019) tested biochars from four feedstocks, each produced at three temperatures, to find minimal $NO_3^-$ sorption and even $NO_3^-$ release. Similarly, Sanford et al. (2019) found that five biochars from diverse

feedstocks and production temperatures had zero $NO_3^-$ binding capacity. Little to no $NO_3^-$ sorption capacity has been commonly observed for biochars produced from a broad range of feedstocks, productions methods, and temperatures (Gai et al., 2014; Hale et al., 2013; Hollister et al., 2013; Li et al., 2018; Wang et al., 2017; Zeng et al., 2013). Though exceptions have been observed in which biochars exhibited high $NO_3^-$ binding capacities (Ahmadvand et al., 2018; Chandra et al., 2020), a recent study determined the average published maximum adsorption capacity ($Q_{max}$) of unmodified biochar for $NO_3$-N to be as low as 1.95 mg $g^{-1}$ (Zhang et al., 2020).

This same study determined the average published $Q_{max}$ of unmodified biochar for $NH_4^+$-N to be 11.19 mg $g^{-1}$ (Zhang et al., 2020). Higher $Q_{max}$ values for biochar and $NH_4^+$ are to be expected, as $NH_4^+$ exists in the cationic form in aqueous environments and would more readily adsorb to negatively charged biochar surfaces. While this theoretical electrostatic affinity is supported by higher $Q_{max}$ values throughout published sorption experiments, inconsistencies can still be found. $Q_{max}$ values lower than 2 mg $NH_4^+$ $g^{-1}$ are commonly observed, for biochars produced from a broad range of temperatures and feedstocks (Hale et al., 2013; Paramashivam et al., 2016; Song et al., 2019; Tian et al., 2016; Uttran et al., 2018; Wang et al., 2015a; Yin et al., 2019; Zhang et al., 2017). While most reported $Q_{max}$ values are less than 20 mg $NH_4^+$-N $g^{-1}$ (Zhang et al., 2020), higher values have been observed (Gao et al., 2015; Yin et al., 2018) Biochars exhibit a broad range of $NH_4^+$ sorption capacities and conflicting trends have emerged. Multiple authors have observed that sorption capacity decreases with increasing production temperature (Gai et al., 2014; Gao et al., 2015; Yin et al., 2018). Lower temperatures have been correlated with higher cation exchange capacity (CEC) (Gai et al., 2014), higher O/C ratios (Yang et al., 2017), and more abundant surface functional groups (Yin et al., 2018). These properties may contribute to biochars with enhanced ability to remove $NH_4^+$ from solution, as they provide a greater number of exchange sites and oxygen-containing functional groups which can react with $NH_4^+$ (Yang et al., 2017). The reverse trend has also been observed, however, with authors noting that an increase in production temperature resulted in higher $NH_4^+$ $Q_{max}$ values (Chandra et al., 2020; Zeng et al., 2013; Zheng et al., 2013). Authors point towards the higher specific surface area (SA) of biochar at higher production temperatures as a critical parameter to predicting $NH_4^+$ adsorption.

Chemical bonding and electrostatic interactions may not be the only mechanism by which biochar retains $NO_3^-$ and $NH_4^+$ in soils. Despite the lack of chemical affinity between $NO_3^-$ and biochar, studies frequently demonstrate the ability of biochar to inhibit $NO_3^-$ leaching in soil column studies and pot trials (Haider et al., 2016; Kameyama et al., 2012; Pratiwi et al., 2016; Yao et al., 2012). While some authors hypothesize the mechanism to be microbial immobilization (Bu et al., 2017), others have found the addition of biochar to stimulate N mineralization (Teutscherova et al., 2018). In addition to chemical and microbial mechanisms, biochar may retain N through physical means (Clough and Condron, 2010). A literature review determined that biochar decreased soil bulk density by 3 to 31% and increased porosity by 14 to 64% (Blanco-Canqui, 2017). Biochar can also alter mean pore size and pore architecture, thereby influencing tortuosity and the residence time of water and nutrients within the soil profile (Lim et al., 2016; Quin et al., 2014). The impact of biochar on hydraulic conductivity appears dependent on soil texture, which highly influences pore structure. While exceptions have been observed, biochar has largely

been shown to decrease the ability of a saturated soil to transmit water (saturated hydraulic conductivity ($K_{sat}$)) in coarse textured soils and increase $K_{sat}$ in finer soils (Blanco-Canqui, 2017). The impact of biochar on these soil physical properties may influence $NO_3^-$ retention through a mechanism known as "nitrate capture," in which $NO_3^-$ molecules become physically entrapped within biochar pores (Haider et al., 2016), potentially leading to increased residence time in crop rooting zones and a greater opportunity for plant uptake (Haider et al., 2020; Kameyama et al., 2012; Kammann et al., 2015).

In this project, biochar characterization, sorption, and soil column experiments were carried out using a robust matrix of commercially-available biochars, produced from diverse feedstocks and at multiple temperatures. The suite of experiments was chosen in order to elucidate the degree to which these biochars: 1) chemically bind $NO_3^-$ and $NH_4^+$; 2) physically alter soil properties which influence saturated hydraulic conductivity; or 3) influence nutrient leaching, through either chemical or physical processes. This information was used to determine the biochar parameters that may optimize hydrologic and nutrient retention benefits in agricultural soils, and to investigate the combination of chemical and physical mechanisms by which these benefits are delivered. Adding to the novelty of this project is that the same soils and biochars were used as those in ongoing 3-year field trials, so that mechanistic laboratory studies can be linked with effects observed in on-farm cropping systems. Results are intended to inform the production or modification of biochar for the delivery of agronomic benefits, as well as to improve predictions on the behaviour of biochar in specific agricultural conditions.

## 2 Materials and methods

### 2.1 Biochar characterization

Seven biochars were obtained from the following feedstocks and produced at the following temperatures: almond shell at 500 °C (AS500, produced by Karr Group Co.), almond shell at 800 °C (AS800, Premier Mushroom and Community Power Co), coconut shell at 650 °C (CS650, Cool Planet), softwood at 500 °C (SW500, Karr Group Co.), softwood at 650 °C (SW650, Cool Planet), and softwood at 800 °C (SW800, Pacific Biochar), and an additional softwood biochar produced at 500 °C and inoculated with a proprietary, yet commercially available, microbial formula (SW500-I, Karr Group Co.). Unless otherwise stated, biochars were sieved to 2 mm and characterized using procedures recommended by the International Biochar Initiative (IBI, 2015): pH and electrical conductivity (EC) were measured at a 1:20 biochar to 18.2 MΩ-cm water (Barnstead Nanopore, Thermo Fisher) dilution (w:v) after solutions were shaken for 90 minutes; total carbon, nitrogen, hydrogen, and oxygen were measured using a dry combustion-elemental analyzer (Costech ECS4010); and moisture, volatile, and ash content were measured as a percent of total dry weight through sequential shifts in furnace temperature (briefly, 2 h at 105 °C, 6 min at 950 °C, and 6 h at 750 °C, respectively) (ASTM D 1762-84, 2011). Particle size distribution was measured by laser diffraction (Coulter LS230). CEC was measured using a combination of the modified ammonium acetate compulsory displacement method (Gaskin et al., 2008) and the rapid saturation method (Mukome et al., 2013; Mulvaney et al., 2004): 0.25g of biochar

was leached with 18.2 MΩ-cm water (w:v) under vacuum (-20 to -40 kPa). Leachate was stored and analysed for dissolved organic carbon (DOC) through combustion (Shimadzu TOC-V). Biochar samples were then washed with 1 M sodium acetate (pH 8.2) until the EC of the elute was the same as the eluant. Samples were rinsed three times with 10 ml of 2-proponal, then dried under vacuum for 10 minutes. To displace sodium ions, biochars were washed with 1 M ammonium acetate of the same volume as was required of sodium acetate. Leachate was collected and analysed for sodium concentration through atomic absorption spectroscopy (Perkin Elmer AAnalyst 800). The micropore specific surface area ($SSA_{\mu p}$) was determined from $CO_2$ adsorption isotherms at 273 K using the Non-Local Density Functional Theory (NLDFT) (Particle Testing Authority, Micromeritics TriStar II Plus 3.0, NLDFT model mod11.df2). Prior to analysis, samples were degassed with $N_2$ at 393K for 16 h.

Fourier transform infrared (FTIR) spectra of AS500, AS800, and SW500 biochars were collected using the diffuse reflectance infrared Fourier transform sampling mode (DRIFT; PIKE Technologies EasiDiff) with air dried samples diluted to 3% with potassium bromide. All FTIR spectra were collected using a Thermo Nicolet 6700 FTIR spectrometer (Thermo Scientific) using 256 scans, 4 cm$^{-1}$ resolution, and a DTGS detector. FTIR bands were assigned as in Parikh et al. (2014). The PZC of AS500, AS800, and SW500 was estimated as the pH at which the zeta potential (ZP) was approximately zero, utilizing a ZetaPlus (Brookhaven Instruments Corp., Holtsville, NY) following the method established in Wang et al. (2016). Briefly, suspensions were prepared by adding 50 ml of 1 mM KCl to 0.1 g of biochar. Samples were sonicated for 1 h, and the pH was adjusted before ZP measurements by adding HCl or KOH dropwise. Ten measurements were taken for each sample. Gross morphological differences among AS500, AS800, and SW500 were visualized by X-ray micro-computed tomography (X-ray microCT) at the Lawrence Berkeley National Laboratory Advance Light Source on beamline 8.3.2, using a beam energy of 21 KeV. Biochars were sieved to 2 mm and mounted in syringes of 8.3 mm diameter for imaging. A total of 1025 projections were acquired using continuous tomography mode with a 4x objective, for a final pixel size of 1.7 µm. Images were reconstructed using Gridrec methods via TomoPy and Xi-CAM (Gürsoy et al., 2014; Pandolfi et al., 2018). Image analysis was completed in Dragonfly, a 3D image analysis software free for non-commercial use (Object Research Systems, Canada).

## 2.2 Soil characterization

Hanford sandy loam (HSL) and Yolo silt loam (YSiL) soils were chosen for continuity between laboratory experiments and ongoing 3-year field trials utilizing the same biochars and soils. Collectively, these soils represent over 260,000 hectares of arable land in California and offer textural distinctions within a range of soils commonly farmed in the Central Valley of California (Soil Survey Staff, 2014). Soils were located via Web Soil Survey (http://websoilsurvey.sc.egov.usda.gov/) and collected from the top 30 cm in fallowed agricultural fields in Parlier, California (HSL) and Davis, California (YSiL). Soils were homogenized and sieved to 2 mm for characterization and column experiments. Colorimetric $NO_3^-$ and $NH_4^+$

measurements were made according to Doane and Horwath (2003) and Verdouw et al. (1978) (Shimadzu UV-1280).
Extractable P was measured using the Olsen sodium bicarbonate extraction (Watanabe and Olsen, 1965). Concentrations of potassium, calcium, magnesium, and sodium were measured by extracting 4 g of soil with 40 ml of 1 M ammonium acetate on a shaker for 30 minutes. Nutrient concentrations of filtered extracts were determined through atomic absorption spectroscopy (Perkin Elmer AAnalyst 800). Total porosity was calculated from the pore volume divided by the total soil volume in representative cores. Pore volume was determined from the difference in weight between saturated and oven-dried (105 °C for 24 h) cores. The pH and EC of soils with and without biochar were measured via 1:2 soil to 18.2 MΩ-cm water (w:v) dilution, after 15 minutes on the shaker and 60 minutes at rest (Thomas, 1996). Soil texture analysis was performed by the Analytical Lab at the University of California, Davis (Davis, CA, USA) using the hydrometer method (Sheldrick and Wang, 1993).

## 2.3 Sorption experiments

To investigate the ability of biochar to adsorb $NH_4^+$ and $NO_3^-$, 0.1 g of biochar was added to 40 ml of solution containing either 0, 50, 100, 200, 400, or 600 mg $L^{-1}$ of $NO_3^-$ (as $KNO_3$) or $NH_4^+$ (as $NH_4Cl$), along with method blanks. All solutions were prepared in 0.1 mM NaCl and, as in Hale et al. (2013a), spiked at 1% volume with a stock solution of 20 g $L^{-1}$ sodium azide to inhibit microbial growth. Monovalent NaCl was chosen to avoid cation bridging reactions during the experiment. All sorption experiments were performed in triplicate at $22 \pm 1$ °C. Tubes were placed on an end-over shaker at 8 rpm for 24 h. Supernatants were passed through a 0.45 µm filter and analysed for colorimetric $NO_3^-$ and $NH_4^+$ (Shimadzu UV-1280) (Doane and Horwath, 2003; Verdouw et al., 1978). Single point sorbed ion concentration was determined at initial concentrations of 100 mg $NO_3^-$ or $NH_4^+$ $g^{-1}$ biochar using Eq. (1).

$$q = \frac{C_0 V_0 - C_f V_f}{m}$$  (1)

Where q is the sorbed ion concentration (mg $g^{-1}$), $C_0$ and $C_f$ are the initial and final sorbate concentrations, respectively (mg $L^{-1}$), $V_0$ and $V_f$ are the initial and final solution volumes, respectively (L), and m is the mass of biochar (g). Langmuir, Freundlich, and Langmuir-Freundlich equations were tested to model the adsorption isotherms, with the Freundlich equation (Eq. (2)) demonstrating the best fit based on $r^2$ values.

$$q = K_f C_f^{\frac{1}{n}}$$  (2)

Where q and $C_f$ are the same as in equation 1, $K_f$ is the Freundlich constant (mg g$^{-1}$), and $1/n$ is the degree of nonlinearity of the isotherm. Excel was used to determine the parameters for the equations. Using batch sorption results, AS500, AS800, and SW500 were selected for further experimentation.

### 2.4 Column experiments

To investigate the influence of biochar on saturated hydraulic conductivity ($K_{sat}$), constant head column experiments were performed in five replicates using the 5 station Chameleon Kit (Soilmoisture Equipment Corporation (SEC) 2816GX). SEC tempe cells, each with a volume of 136.4 cm$^3$, were packed with soils amended with 0 and 2% (w/w) AS500, AS800, or SW500 biochars, to a bulk density of $1.34 \pm 0.02$ g cm$^{-3}$. Soils and biochars were thoroughly and homogenously mixed prior to being added to tempe cells, and packed using the dry method according to Gibert et al. (2014). An application rate of 2% was chosen as the midrange of those represented in similar experiments (Blanco-Canqui, 2017), and is within recommended ranges for field application (Guo, 2020; Jeffery et al., 2011). Columns were saturated for 24 h before the start of each experiment. Each column was gravity-fed a solution of 0.1 mM CaCl$_2$ at a pressure head of 34 cm for 10 pore volumes. Divalent CaCl$_2$ was chosen to avoid dispersion and the creation of preferential flow pathways. $K_{sat}$ was calculated using data produced by SEC pressure transducers and PressureLogger software, which monitored pressure head and flow over time. Columns were also used to investigate nutrient retention and leaching in the HSL amended with 0 and 2% biochar. $K_{sat}$ trials with the YSiL demonstrated that flow rates were very low (~0.044 cm s$^{-1}$), creating logistical challenges for investigating nutrient retention and leaching in this soil. Additionally, the impact of NO$_3^-$ leaching is more pronounced in coarsely textured soils. Thus, leaching experiments were conducted in HSL columns only. To remove existing nitrogen, columns were flushed for 10 pore volumes with 0.1 mM CaCl$_2$, after which 50 mg L$^{-1}$ of both NO$_3^-$ and NH$_4^+$ (as NH$_4$Cl and KNO$_3$) was gravity-fed through columns for 15 pore volumes. Leachate was collected every 0.5 pore volumes and analysed for colorimetric NO$_3^-$ and NH$_4^+$ as in sorption experiments (Doane and Horwath, 2003; Verdouw et al., 1978).

### 2.5 Statistical analysis

All data were analysed with linear models (lm(response variable ~ biochar)) and one-way analysis of variance (ANOVA) in the stats and Tidyverse packages in R (R Core Team, 2020; Wickham et al., 2019). When more than one soil type was tested (as in $K_{sat}$ measurements), separate models were built for each soil type to determine the effect of biochar within soil types. For analysis of results, all effects with p-values < 0.05 were considered significant. P-values were generated using the emmeans package in R (Lenth, 2019) and corrected for multiple comparisons using Tukey's honestly significant difference (HSD) method. Plots were generated in R using the ggplot2 package (Wickham, 2016) and visualized as the mean plus or minus the standard error of the means.

# 3 Results

## 3.1 Biochar characterization

Biochars exhibited a broad range of chemical and physical properties depending on their production temperature and feedstock
220 (Tables 1 and 2). All biochars contained less than 1% nitrogen, spanning from SW800 at 0.13% to CS650 at 0.79%. Almond
shell biochars contained 4-6x more nitrogen than softwood biochars produced at the same temperature. Softwood biochars
produced at 500 and 800 °C had substantially higher $SSA_{\mu p}$ than almond shell biochars produced at the same temperatures. It
should be noted, however, that $SSA_{\mu p}$ measured by $CO_2$ adsorption frequently results in higher values than surface area
measured by $N_2$, as $CO_2$ can access micropores unavailable to $N_2$ (Maziarka et al., 2021; Zeng et al., 2013). While results from
225 each method tend to be well correlated and are considered to provide complementary information (Sigmund et al., 2017),
neither should not be regarded as providing precise total surface area. Overall, AS800 possessed the most unique properties,
with the lowest carbon content at 35.3%, the highest ash content at 55.4%, the highest EC at 27.2 mS cm$^{-1}$, a basic pH of
10.13, the highest O/C ratio at 0.56, and the second highest CEC at 53.77 cmol$_c$ kg$^{-1}$

230 **Table 1: Select chemical and physical biochar properties (n=3) ± standard error of the means**

|  | AS500 | AS800 | CS650 | SW500 | SW500-I | SW650 | SW800 |
|---|---|---|---|---|---|---|---|
| Carbon (%) | 65.8 ± 0.5 | 35.3 ± 0.3 | 71.2 ± 0.7 | 70.9 ± 0.3 | 63.5 ± 0.3 | 78.3 ± 0.4 | 41.8 ± 0.5 |
| Nitrogen (%) | 0.76 ± 0.01 | 0.55 ± 0.02 | 0.79 ± 0.04 | 0.13 ± 0.03 | 0.69 ± 0.01 | 0.29 ± 0.01 | 0.13 ± 0.03 |
| Oxygen (%) | 17.1 ± 0.8 | 26.4 ± 0.8 | 13.7 ± 0.6 | 17.1 ± 0.6 | 20.1 ± 0.2 | 10.2 ± 0.2 | 15.3 ± 0.9 |
| Hydrogen (%) | 3.1 ± 0.04 | 1.8 ± 0.02 | 3.2 ± 0.06 | 3.8 ± 0.01 | 3.8 ± 0.03 | 2.9 ± 0.07 | 1.5 ± 0.05 |
| Molar O/C ratio | 0.19 ± 0.01 | 0.56 ± 0.01 | 0.15 ± 0.01 | 0.18 ± 0.01 | 0.24 ± 0 | 0.1 ± 0 | 0.27 ± 0.01 |
| Molar H/C ratio | 0.55 ± 0.01 | 0.62 ± 0.01 | 0.54 ± 0.01 | 0.63 ± 0 | 0.71 ± 0.01 | 0.44 ± 0.01 | 0.42 ± 0.02 |
| Volatile (%) | 30.7 ± 2.7 | 28.2 ± 0.5 | 32.1 ± 0.4 | 38.0 ± 0.9 | 38.8 ± 1.2 | 26.9 ± 0.3 | 21.7 ± 0.2 |
| Ash (%) | 19.0 ± 1.0 | 5545 ± 0.8 | 5.3 ± 0.2 | 4.5 ± 0.1 | 9.2 ± 0.5 | 4.5 ± 0.3 | 31.5 ± 1.2 |
| pH | 9.3 ± 0.02 | 10.1 ± 0.01 | 7.8 ± 0.02 | 7.9 ± 0.02 | 10.4 ± 0.01 | 8.0 ± 0.03 | 10.3 ± 0.01 |
| EC (mS cm$^{-1}$) | 3.2 ± 0.01 | 27.2 ± 0.1 | 0.3 ± 0 | 2.5 ± 0.02 | 2.1 ± 0.02 | 0.1 ± 0 | 2.1 ± 0.01 |
| DOC (mg kg$^{-1}$) | 38322.1 ± 1776.6 | 1055.9 ± 52.9 | 644.5 ± 77.1 | 43776.2 ± 1103.8 | 32171.2 ± 934.8 | 423.4 ± 50.6 | 475.2 ± 66.9 |
| CEC (cmol$_c$ kg$^{-1}$) | 24.0 ± 0.6 | 52.7 ± 0.8 | 26.8 ± 1.1 | 16.5 ± 0.4 | 34.1 ± 0.2 | 21.7 ± 0.4 | 60.8 ± 0.8 |
| Mean particle size (μm) | 464.0 | 269.8 | 609.1 | 493.6 | 241.1 | 212.3 | 139.4 |
| Median particle size (μm) | 590.6 | 334.8 | 931.2 | 763.5 | 312.8 | 446.3 | 171.2 |
| $SSA_{\mu p}$ (m$^2$ g$^{-1}$) | 54.7 | 188.2 | 233.6 | 93.5 | 152.6 | 305.6 | 363.6 |

**EC = electrical conductivity; DOC = dissolved organic carbon; CEC = cation exchange capacity; ; $SSA_{\mu p}$ = micropore specific surface area**

The IR spectra of AS500 and SW500 contained carboxyl and aromatic functional groups present at 1697 and 1703 cm$^{-1}$ (C=O) and 1410 and 1418 cm$^{-1}$ (COO$^-$); aromatic bands around 1580 cm$^{-1}$; C=C skeletal vibrations; out of plane C-H bending vibrations (700 to 900 cm$^{-1}$) associated with adjacent aromatic hydrogen bonds; and aromatic C=C and C=O stretching vibrations (1581 and 1589 cm$^{-1}$) (Fig. 1a, Table S1). AS800 spectra contained a strong band at 1405 cm$^{-1}$ representing substantial contributions of COO$^-$, and multiple sharp IR peaks from ~1000 to 700 cm$^{-1}$ arising from metal oxide vibrations (Fig. 1a, Table S1). The high contribution of O-rich functional groups and metal oxide vibrations is consistent with the elemental analysis of AS800, which showed high oxygen and ash content (Table 1). The measured PZC for each of the three tested biochars is as follows: 3.2 for AS500, 6.8 for AS800, and 3.9 for SW500. The higher PZC of AS800 is consistent with the higher ash and metal-oxide content previously described. Each biochar was visually distinct at the macroscale (Fig. 1b). Animated reconstructions of biochar particles are provided in the supplementary information (SI) (Fig. S1a, S1b, and S1c). The macro-pores (>50 µm) of SW500 were more uniform in size compared to those of AS500 and AS800 (Fig. 1b, S1a, S1b, and S1c). The softwood chips added to the AS500 feedstock matrix (at 25% w/w to assist with pyrolysis) are visible in the background, and contrast sharply with the almond shells (Fig. 1b and S1a). The macro-pores of AS800 appeared to increase in size (most visible in the bottom right of AS800 Fig. 1a, and in the animated reconstruction in figure S1b), due to the collapse of the lacy carbon pores that were visible in AS500 (Fig. 1b and S1a). The increase in production temperature resulted in more binomial pore size distribution in AS800, with larger macropores and an increased quantity of micropores, as observed by X-ray microCT and $CO_2$ SSA$_{\mu p}$ measurements (Table 1, Fig. 1b, SI Fig. S1b), leading to an overall increase in total surface area.

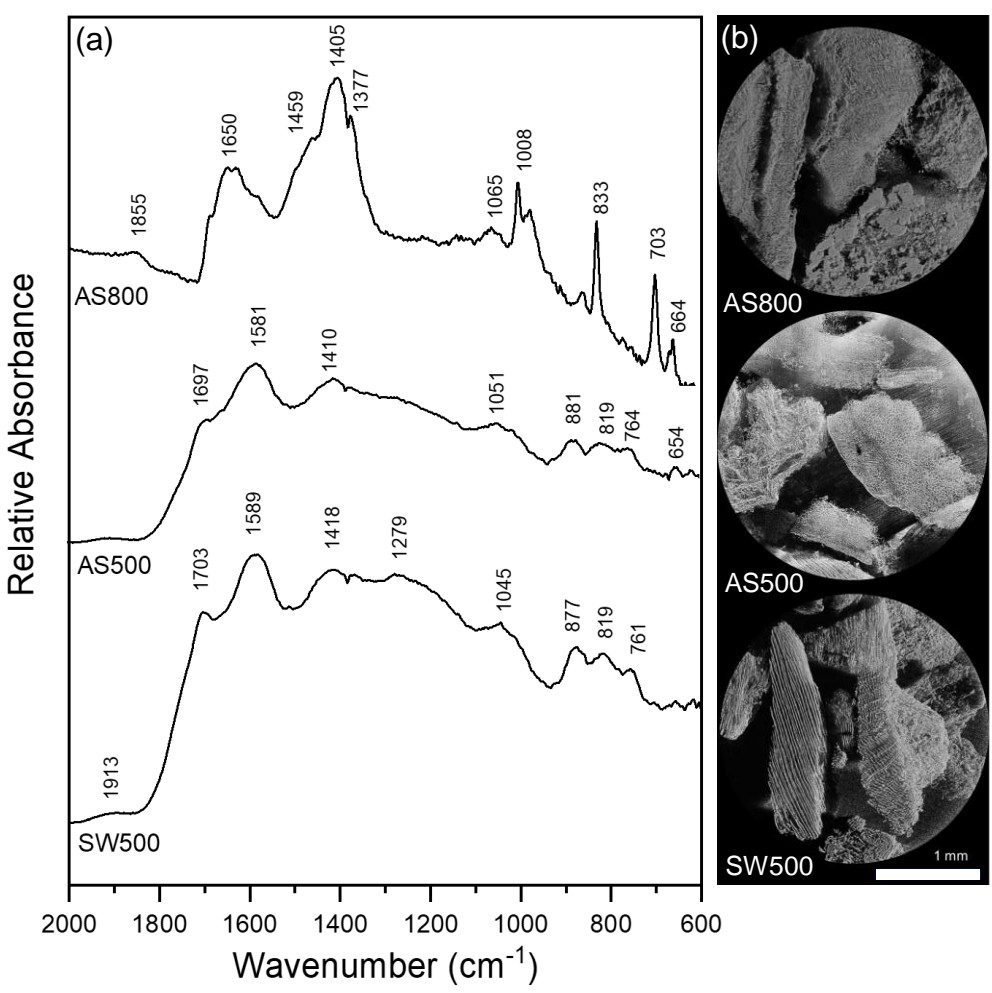

**Figure 1: a) DRIFT spectra of AS800, AS500, and SW500 biochars. Samples diluted with potassium bromide to 3% sample, and collected with 256 cm⁻¹ scans with a 4 cm⁻¹ resolution; b) X-ray microCT images of AS800, AS500, and SW500 biochars.**


### 3.2 Soil characterization

Table 3 contains select chemical and physical properties of soils used in this study. The finer textured YSiL had a porosity of 42.5%, a sand content of 24%, and a clay content of 33%, compared to the coarser HSL with a porosity of 29.9%, and sand 260 and clay contents of 59%, and 12%, respectively. Both HSL and YSiL contained substantial levels of $NO_3^-$, calcium, magnesium, and potassium, and were slightly above neutral at a pH of 7.3.

**Table 3: Select physical and chemical properties of Hanford Sandy Loam (HSL) and Yolo Silt Loam (YSiL) (n=3) ± standard error of the means**

|  | HSL | YSiL |
|---|---|---|
| $NH_4^+$ (mg kg$^{-1}$) | $0.74 \pm 0.05$ | $1.02 \pm 0.14$ |
| $NO_3^-$ (mg kg$^{-1}$) | $34.49 \pm 0.50$ | $40.40 \pm 1.05$ |
| Ca (mg kg$^{-1}$) | $943.41 \pm 11.56$ | $2191.26 \pm 7.19$ |
| Mg (mg kg$^{-1}$) | $58.05 \pm 1.62$ | $508.50 \pm 11.60$ |
| K (mg kg$^{-1}$) | $55.91 \pm 0.99$ | $360.05 \pm 0.70$ |
| Na (mg kg$^{-1}$) | $118.09 \pm 2.27$ | $146.56 \pm 0.73$ |
| Olsen P (mg kg$^{-1}$) | $9.19 \pm 0.12$ | $9.83 \pm 0.15$ |
| pH | $7.3 \pm 0.09$ | $7.3 \pm 0.05$ |
| EC (µs cm$^{-1}$) | $427.33 \pm 2.84$ | $269.25 \pm 1.92$ |
| Porosity (%) | $29.9 \pm 0.35$ | $42.5 \pm 0.42$ |
| Sand (%) | $59.0 \pm 1.4$ | $24.0 \pm 0.9$ |
| Clay (%) | $12.0 \pm 0.9$ | $33.0 \pm 0.5$ |


EC = electrical conductivity

### 3.3 Sorption

All biochars exhibited the capacity to remove $NH_4^+$ from solution (Fig. 2), though $K_f$ values were low (Table 4). Single point
concentration tests at a $C_0$ of 100 mg L$^{-1}$ revealed the following hierarchy of sorption capacities, in order of lowest to highest:
SW650 < SW500 < CS650 < SW500-I < AS500 < SW800 < AS800 (Table 4). These q values spanned 0.70 (SW650) to 7.15
(AS800) mg g$^{-1}$, or removal efficiencies of 0.70 and 7.15%. AS800 exhibited the greatest $K_f$ value at 0.16 mg $NH_4^+$ g$^{-1}$. Only
AS500 exhibited the ability to remove $NO_3^-$ from solution. The other six biochars released, rather than removed, $NO_3^-$ (Fig.
S1). For AS500, the single point concentration test at a $C_0$ of 100 mg L$^{-1}$ revealed a removal efficiency of 1.74%, or a q of 1.74
mg g$^{-1}$ (Table 4). All tested models were poor fits for the AS500 and $NO_3^-$ isotherm, including the Freundlich equation with an
$r^2$ of 0.57. As such, $K_f$ and 1/n values provided in Table 4 should be regarded with caution.

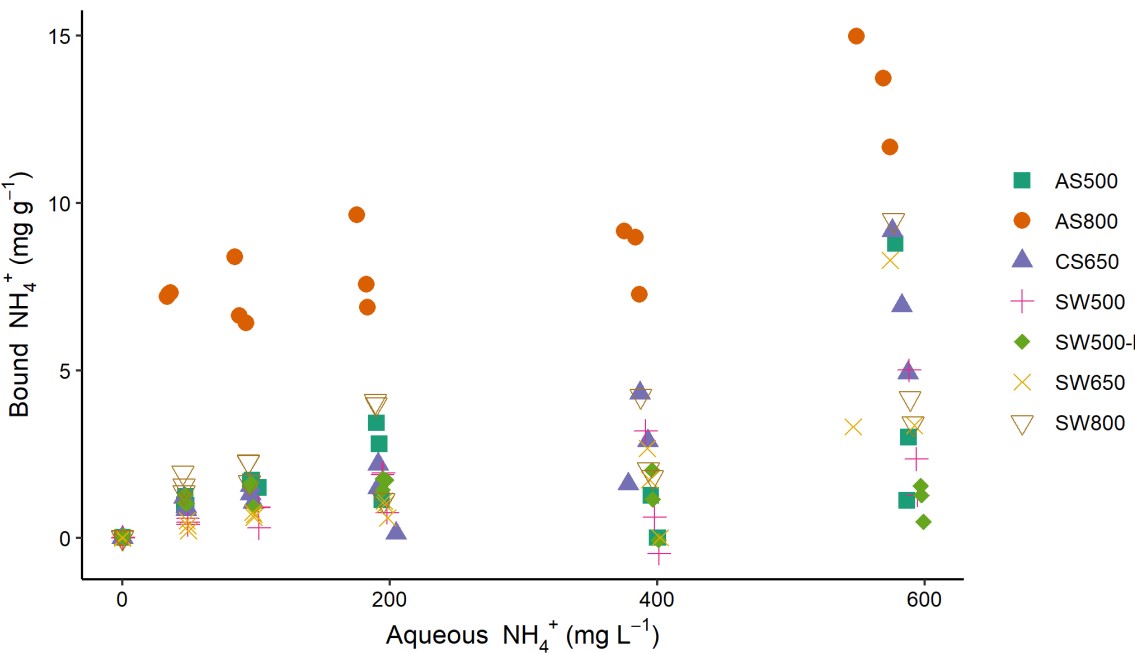

 **Figure 2. Sorption isotherms for ammonium and biochars, performed in at 22 ± 1 °C. All solutions were prepared in 0.1 mM NaCl and spiked at 1% volume with a stock solution of 20 g L$^{-1}$ sodium azide to inhibit microbial growth.**




**Table 4: Concentration of ions bound to biochars (mg NH$_4^+$ or NO$_3^-$ g$^{-1}$) at single point concentration of 100 mg L$^{-1}$, and Freundlich model parameters (n=3). Nitrate parameters reported for only one biochar (AS500), as all other biochars released rather than removed NO$_3^-$.**

| | Single point concentration | | Freundlich parameters | | |
| --- | --- | --- | --- | --- | --- |
| Biochar | q (mg NH$_4^+$ g$^{-1}$) | standard error | 1/n | K$_f$ (mg NH$_4^+$ g$^{-1}$) | r$^2$ |
| AS500 | 1.63 | 0.05 | 0.71 | 0.05 | 0.90 |
| AS800 | 7.15 | 0.51 | 0.77 | 0.16 | 0.84 |
| CS650 | 1.30 | 0.12 | 0.65 | 0.06 | 0.75 |
| SW500 | 0.70 | 0.17 | 0.83 | 0.01 | 0.91 |
| SW500-I | 1.37 | 0.18 | 0.52 | 0.08 | 0.73 |
| SW650 | 0.69 | 0.03 | 0.68 | 0.03 | 0.89 |
| SW800 | 2.06 | 0.17 | 0.77 | 0.04 | 0.90 |
| Biochar | q (mg NO$_3^-$ g$^{-1}$) | standard error | 1/n | K$_f$ (mg NO$_3^-$ g$^{-1}$) | r$^2$ |
| AS500 | 1.74 | 0.47 | 0.49 | 0.22 | 0.57 |


### 3.4 Soil columns- hydraulic conductivity and breakthrough curves

There was a significant effect of biochar (p = 0.001) on saturated hydraulic conductivity in both soils. In the HSL, AS500 and SW500 each decreased K$_{sat}$ by 75%, from the control at 1.2 cm s$^{-1}$ to 0.3 cm s$^{-1}$ (p = 0.023) (Fig. 3). AS800 caused a 12.5% decrease in K$_{sat}$ to 1.05 cm s$^{-1}$, though the effect was not significant (p = 0.939). In the YSiL, AS500 decreased K$_{sat}$ by 63.6%, from the control at 0.044 cm s$^{-1}$ to 0.016 (p < 0.001). SW500 caused a decrease of 79.5%, to 0.009 cm s$^{-1}$ (p < 0.001). In contrast to its effect on HSL, AS800 increased K$_{sat}$ in YSiL by 97.7%, to 0.087 cm s$^{-1}$ (p < 0.001).

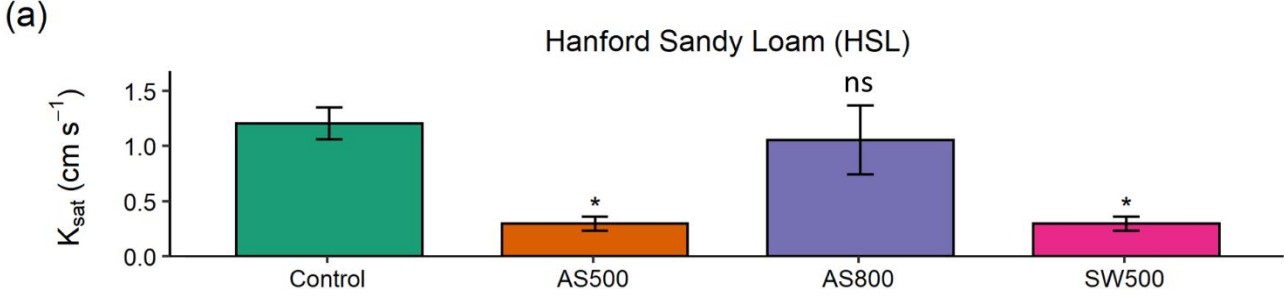

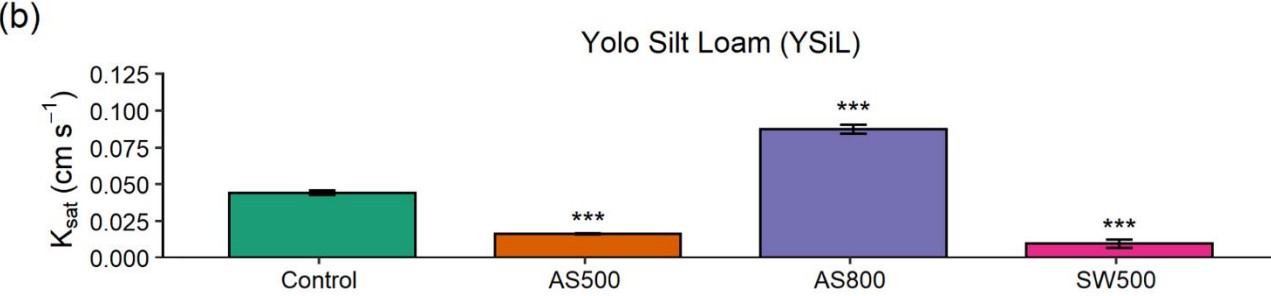

**Figure 3: Impact of 0 and 2% addition of AS500, AS800, and SW500 biochars on saturated hydraulic conductivity ($K_{sat}$) in a) a**

**Hanford Sandy Loam (HSL) soil and b) a Yolo Silt Loam (YSiL) soil (n=5). Symbols denote significance levels as follows: ns = not significant, \*p<0.05, \*\*p<0.01, \*\*\*p<0.001. P-values refer to comparisons between treatments and the control within each pore volume, and were corrected for multiple comparisons using Tukey's HSD method.**

Figure 4 illustrates the $NH_4^+$ and $NO_3^-$ breakthrough curves for HSL amended with 0 and 2% AS500, AS800, and SW500.

Biochar affected the timing and quantity of $NH_4^+$ (introduced in pore volumes 11-25 at 50 mg $L^{-1}$) leached from the soil column (Fig. 4a). The estimated breakthrough point, or the pore volume at which the concentration of the leachate equals 0.5x the concentration of the incoming solution ($C/C_0 = 0.5$), was reached as follows, in order of fastest to slowest for $NH_4^+$: HSL (control) at pore volume 14.3, SW500 at 15.5, AS500 at 16.2, and AS800 at 18.1. Biochar also significantly decreased the total amount of $NH_4^+$ in the leachate at all pore volumes, as follows, in order of least to most retention: HSL < SW500 < AS500

< AS800 (Fig. 5a). At pore volume 15, AS500 decreased the $NH_4^+$ concentration of the leachate compared to the control (HSL = 37.33 mg $L^{-1}$) by 30.5% (p < 0.001), AS800 by 78.1% (p < 0.001), and SW500 by 24.4% (p = 0.002). This effect was diminished by pore volume 25, where differences from the control (HSL= 41.69 mg $L^{-1}$) were decreased to 21.8% by AS500 (p < 0.001), 28.9% by AS800 (p < 0.001), and 8.5% by SW500 (not statistically significant at p = 0.463).

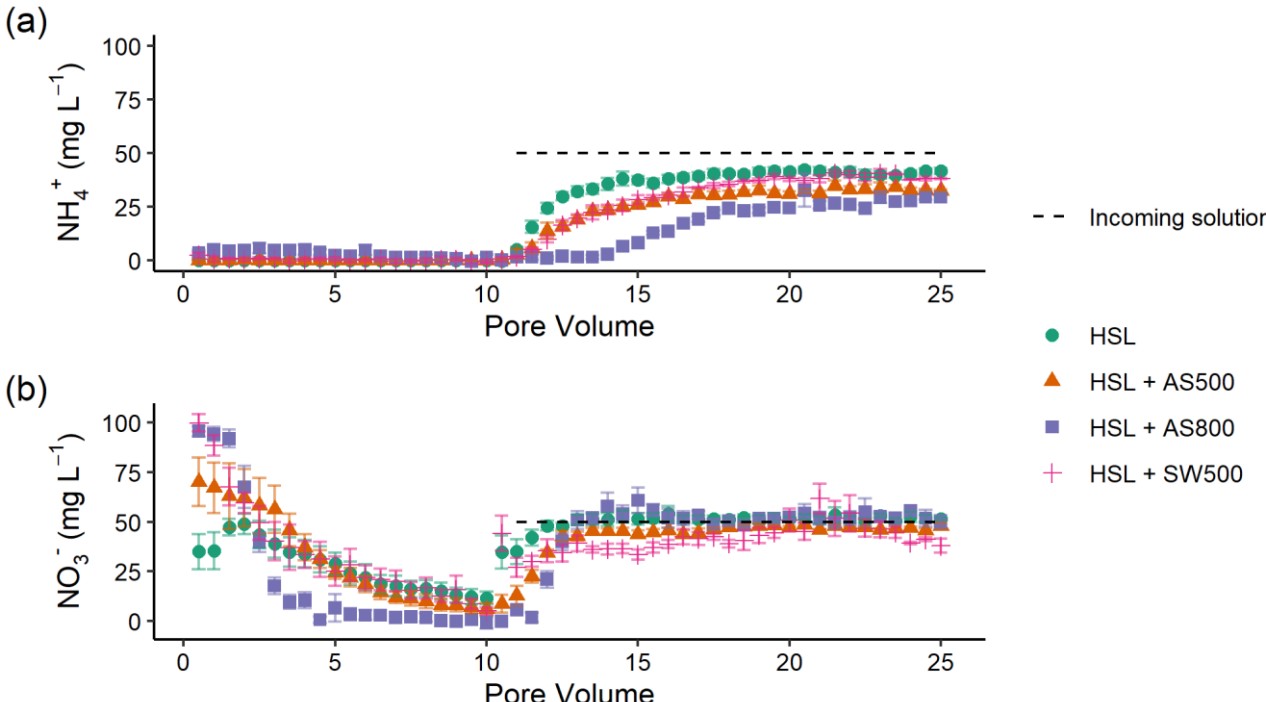

**Figure 4: Breakthrough curves for a) ammonium and b) nitrate in a Hanford Sandy Loam (HSL) soil with 0 and 2% additions of AS500, AS800, and SW500 biochars. Native soil nitrogen was flushed in pore volumes 0-10 with a 0.1 mM CaCl$_2$ solution, after which 50 mg L$^{-1}$ solutions of NH$_4^+$ and NO$_3^-$ were gravity-fed through soil columns (n=5). Error bars represent standard error of the means.**

Estimated NO$_3^-$ breakthrough points for biochar amended soils were each within 0.5 pore volumes of the control (pore volume 11.4), indicating that biochar had little to no effect on the timing of NO$_3^-$ release from HSL. The effect of biochar on the total quantity of nitrate released was also less substantial than for NH$_4^+$ (Fig. 4b). Only SW500 significantly decreased the concentration of NO$_3^-$ in the leachate compared to the control. At pore volume 15, SW500 inhibited NO$_3^-$ transport by 35.01% (p = 0.002) (Fig. 5b). This effect was not present at pore volume 20, and was slightly lessened to 26.5% by pore volume 25 (marginally significant at p = 0.098).

Evaluation of data revealed no evidence of preferential flow in any replicate in any of the columns, which were carefully prepared according to the dry packing method in Gibert et al. (2014). This is demonstrated in the small error bars in figures of nutrient concentrations across pore volumes, as well as in hydraulic conductivity measurements taken by data loggers. To further monitor columns for preferential flow, the use of a conservative tracer could be considered in future experiments.

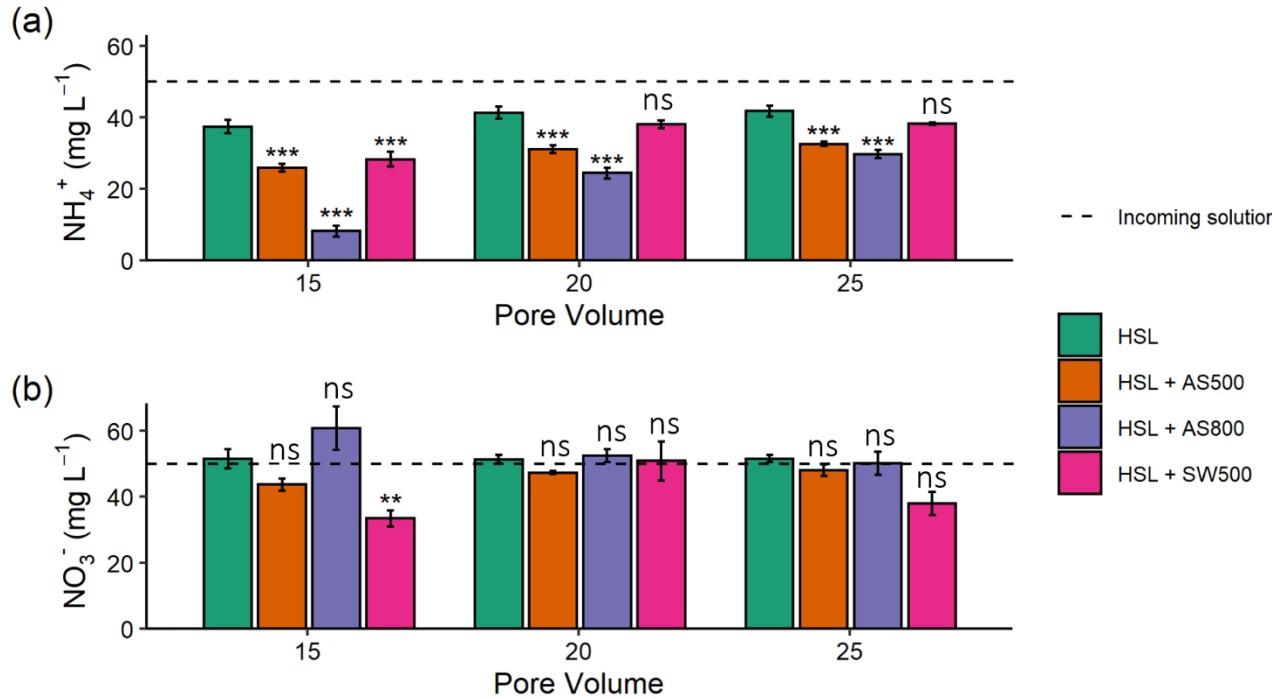

**Figure 5: Quantity of a) ammonium and b) nitrate in Hanford Sandy Loam (HSL) soil columns with 0 and 2% additions of AS500, AS800, and SW500 biochars in pore volumes 15, 20, and 25 (n=5). Error bars represent standard error of the means. Symbols denote significance levels as follows: ns = not significant, \*p<0.05, \*\*p<0.01, \*\*\*p<0.001. P-values refer to comparisons between treatments and the control within each pore volume, and were corrected for multiple comparisons using Tukey's HSD method.**

## 4 Discussion

### 4.1 Biochar properties and nutrient removal

An increase in biochar production temperature was generally associated with higher ash content, pH, EC, and surface area, as well as decreased carbon and hydrogen content, and DOC. These trends are consistent with those of a recent meta-analysis of 533 published datasets (Hassan et al., 2020).Contrary to trends reported in the meta-analysis regarding high temperature biochars, AS800 had a high O/C ratio and CEC (Hassan et al., 2020). The unusual O content of AS800 is attributed to the high ash content, and possibly due to oxidation through exposure to air immediately after gasification while still hot. As expected, the IR spectra of AS500 and SW500 were notably similar, having been produced at the same temperature by the same company via fractional hydropyrolysis. Additionally, the AS500 biochar included 25% softwood chips to aid the pyrolysis process. By

contrast, AS800 was produced via gasification and contained distinct peaks in the IR spectrum. This biochar also performed distinctly different from other biochars in all experiments conducted.

The ability of all seven biochars to retain $NH_4^+$ within the studied concentration ranges is consistent with a recent literature review of 77 studies (Zhang et al., 2020). AS800 exhibited substantially higher $NH_4^+$ binding capacity than the other biochars tested. While it is typical for biochars produced at high temperatures to have low O/C ratios and low CEC (Hassan et al., 2020), AS800 had the largest O/C ratio at 0.56 (presumably due to the high ash content and possible post-pyrolysis oxidation), and the second highest CEC at 52.75 $cmol_c$ $kg^{-1}$. These properties, as well as the $\nu_s(COO^-)$ IR band at 1405 $cm^{-1}$, explain the high

$NH_4^+$ retention, as they indicate increased exchange sites and oxygen-containing functional groups which can react with $NH_4^+$. The relationship between these biochar properties and $NH_4^+$ binding capacity was also demonstrated with SW800, which had the highest CEC at 60.83 $cmol_c$ $kg^{-1}$, the second highest O/C ratio at 0.27, and the second highest $NH_4^+$ binding capacity. Consistent with prior biochar studies (Fidel et al., 2018; Georgiou et al., 2021; Hu et al., 2020; Jing et al., 2019), these data suggest that $NH_4^+$ is bound to biochar through electrostatic interactions. The demonstrated relationship between $NH_4^+$ sorption

and large O/C ratios and CECs is also consistent with prior studies (Gai et al., 2014; Yang et al., 2017). By contrast, no clear trends between surface area (i.e., $SSA_{\mu p}$) and $NH_4^+$ retention emerged in this experiment, as observed in other studies (Chandra et al., 2020; Zeng et al., 2013; Zheng et al., 2013). However, Zeng et al. (2013) did observe that surface area and CEC were not consistently good predictors of $NH_4^+$, and that other factors must be considered.

That six of the seven biochars did not retain $NO_3^-$, and in most cases released $NO_3^-$, is consistent with most published studies (Gai et al., 2014; Hale et al., 2013; Hollister et al., 2013; Li et al., 2018; Sanford et al., 2019; Wang et al., 2017; Zeng et al., 2013; Zhang et al., 2020; Zhou et al., 2019). Electrostatic repulsion is commonly cited as the mechanism, as most biochars contain carboxyl-rich surface functional groups and have PZCs below agronomic soil pH values (Peiris et al., 2019; Uchimiya et al., 2011; Wang et al., 2020). The PZC values obtained for AS500, AS800, and SW500 were indeed each lower than solution

pH, indicating carboxyl functional groups were predominately deprotonated during sorption experiments. Despite a PZC of 3.2, AS500 exhibited minor affinity for $NO_3^-$. Though the data do not provide a clear mechanistic process, the relatively high ash content and low CEC likely facilitated sorption via anionic binding, as positive charges from biochar ash could bind $NO_3^-$. This is consistent with data from prior studies (Wang et al., 2015b). While AS800 and SW800 had higher ash contents, they also had substantially greater CECs. Inhibited binding between the positively charged metals in the ash and the aqueous $NO_3^-$

may be attributed to the electrostatic repulsion from deprotonated surface functional groups (Tan et al., 2020; Tong et al., 2019; Zhang et al., 2020).

## 4.2 Column experiment- nutrient retention

As in sorption trials, AS800 retained the greatest quantity of $NH_4^+$ in column studies, followed by AS500 and SW500. This suggests that the chemical affinity between biochar and $NH_4^+$ is the controlling factor on the flow of $NH_4^+$ through biochar-amended soils, as the order did not change between sorption and column experiments. By contrast, data indicate that the flow of $NO_3^-$ may be dictated, though minor in effect, by physical means (Clough and Condron, 2010). This is consistent with a study which found no evidence of $NO_3^-$ adsorption to corn stalk biochar surfaces, but determined $NO_3^-$ to be physically retained via diffusion into biochar or through interaction within biochar pores (Tong et al., 2019). Inconsistent with our results, however, the corn stalk biochar showed substantial retention of $NO_3^-$, though this study investigated pure biochar without soil, and a biochar produced from a different feedstock. Indeed, biochar feedstock has a profound impact on its porosity, with materials containing higher ash content typically leading to a lower total porosity biochar (Leng et al., 2021). Unlike in sorption trials, AS500 did not retain significant quantities of $NO_3^-$. This suggests a weak chemical affinity between AS500 and $NO_3^-$, in which $NO_3^-$ was readily desorbed from AS500. Complete desorption between biochar and $NO_3^-$ has been previously reported (Hale et al., 2013). SW500, however, significantly inhibited the flow of $NO_3^-$, despite not exhibiting chemical affinity in sorption trials. Wood biomass biochar produced at 400-700 °C has been noted as ideal for producing high porosity biochars due to its low ash content, high lignin content, and preservation of its original pore structure (Leng et al., 2021). Thus, SW500 is predicted to have the highest total porosity of the three biochars used, partly due to macropore contributions as observed via X-ray microCT. This reinforces that nitrate capture likely occurred, in which nitrate retention is facilitated by increased surface area and porosity (Haider et al., 2016, 2020; Kameyama et al., 2012; Kammann et al., 2015).

Indeed, SW500 had a substantially larger $SSA_{\mu p}$ than AS500 (93.5 compared to 54.7 m$^2$ g$^{-1}$). AS800, however, had an even greater $SSA_{\mu p}$ at 188.2 m$^2$ g$^{-1}$, but exhibited no capacity to retain $NO_3^-$, likely due to its high ash content, and the formation of larger macropores pictured in X-ray microCT images (Fig. 1b), which are not probed via the $CO_2$ surface area approach. Larger pores may have allowed water to move through the biochar more quickly and limited the flow of $NO_3^-$ into micropores where it could be retained. This is consistent with the increase in soil $K_{sat}$ after addition of AS800 in YSiL, and the smaller effect of AS800 in HSL compared to AS500 and SW500 (discussed in section 4.3). Future investigation should include measurements of biochar surface area utilizing both $CO_2$ and $N_2$ adsorption. While $CO_2$ is commonly used to probe micropores in carbon-based materials (Maziarka et al., 2021; Sigmund et al., 2017; Zhu et al., 2011), IBI criteria recommends the use of $N_2$ for biochar analysis (International Biochar Initiative, 2015). Including $N_2$ measurements would aid in standardization across studies. Furthermore, the differences in results from each method may be descriptive of the relative pore size distribution between each biochar in this study. Differences in pore size distributions, as observed by X-ray microCT, have been demonstrated to have a varying effect on water retention and conductivity in previous studies (Devereux et al., 2013; Quin et al., 2014). The strong $NH_4^+$ binding capacity and high CEC of AS800 suggests a highly negatively charged surface. Electrostatic repulsion between AS800 and $NO_3^-$, therefore, may have also prevented nitrate capture. Together with sorption

results, these breakthrough curves add to a growing body of literature which suggests that unmodified biochars may have a strong role in decreasing $NH_4^+$ mobility in soils through chemical retention (Gai et al., 2014; Hale et al., 2013; Hollister et al., 2013; Li et al., 2018; Sanford et al., 2019; Wang et al., 2017; Zeng et al., 2013; Zhang et al., 2020; Zhou et al., 2019). Nitrate capture may have a role to play for reducing $NO_3^-$ mobility, but is unlikely to be a substantial force without chemical or physical modification of biochars. Modification of biochar has been shown to increase nutrient retention (Zhang et al., 2020), and

provides a promising opportunity to reduce $NO_3^-$ leaching in agricultural soils.

### 4.3 Column experiment- saturated hydraulic conductivity

AS500 and SW500 decreased $K_{sat}$ by 75% in HSL. AS800 also decreased $K_{sat}$ in HSL, though to a lesser extent and without statistical significance. This effect is in agreement with a literature review of 26 similar studies, which consistently demonstrate decreased $K_{sat}$ in coarse textured soils after biochar amendment (Blanco-Canqui, 2017). This effect is hypothesized to be the

result of increased surface area, microporosity, and tortuosity, which can slow the movement of water through soils. This decrease in $K_{sat}$, along with the prior discussion of SW500 porosity, further explain the retardation of $NO_3^-$ transport in SW500 columns. By contrast, in fine textured soils, biochar typically increases $K_{sat}$ due to decreased bulk density and an increase in total porosity and mean pore size (Blanco-Canqui, 2017). This is consistent with the 98% increase in $K_{sat}$ in YSiL after amendment with AS800, but contrasts with the 64 and 80% reduction after the addition of AS500 and SW500, respectively.

The microCT data shows that AS800 has the most macropores which would permit greater water flow, whereas AS500 and SW500 have more micropores which can inhibit water flow due to matric forces greatly exceeding gravity forces. Though pore size was not quantitatively measured in this study, it is possible that the pores of AS500 and SW500 were small enough to decrease mean pore size in the coarse soil as in Devereux et al. (2013) but were not large or numerous enough to increase $K_{sat}$ in a fine soil. By contrast, the collapse of the lacy carbon pores in the AS500 compared to AS800 lead to the formation of

both additional small pores with greater surface area (confirmed by BET), and larger macro-pores (as visualized by X-ray microCT) in AS800. This may indicate an ability for AS800 to increase macroporosity, mean pore size, and pore connectivity in YSiL, as seen in other studies (Quin et al., 2014). Broadly, the ability of each biochar to substantially influence the movement of water through each soil underscores its effect on the physical composition of soils. This fact contributes to the hypothesis that $NO_3^-$ capture may have occurred in the case of SW500.

**4.4 Implications for field conditions**

Recent meta-analyses have concluded that biochar substantially increased soil water content at field capacity and permanent wilting point, in the field and lab, in coarse textured soils only (Blanco-Canqui, 2017; Razzaghi et al., 2020). Despite these observed trends, benefits have been observed in fine textured soils as well, including reduced crop water stress, increased yield (Kerré et al., 2017; Nawaz et al., 2019), and reduced crop loss during deficit irrigation (Madari et al., 2017). Other authors

have reported little to no effect, or transient effects, of biochar on soil water dynamics in both fine and coarse textured soils

(Jones et al., 2012; McDonald et al., 2019; Nelissen et al., 2015). The results of this study suggest these unmodified biochars may increase the residence time of water in sandy soils and increase drainage in fine textured soils during irrigation or flooding events, or when soils are otherwise saturated. Results also suggest biochar may increase the residence time of $NH_4^+$ in neutral or basic soils. These effects may be particularly relevant for flooded agricultural systems such as rice, where $NH_4^+$ is the

primary source of N and water retention is a key parameter for success (Minami, 1995). Indeed, 95% of California rice production occurs in the Sacramento Valley, where both the YSiL and HSL soils are common (http://rice.ucanr.edu/About_California_Rice/). Data from these trials may help growers in regions with similar soil textures determine if biochar can increase water and nutrient retention in their systems. However, results cannot be extrapolated to dryland agriculture or in soils that experience wet-dry cycles, as unsaturated hydraulic conductivity was not measured. In order

to determine how these biochars may behave in unsaturated conditions, three-year processing tomato field trials are currently underway with the same biochars and soil textures. The intent is to observe the field-scale effects of these biochars on soil-water and nitrogen dynamics.

## 5 Conclusion

This study provides novel contributions to our understanding of biochar in soils by investigating the combination of chemical

and physical mechanisms through which biochar influences nutrient retention and hydraulic conductivity, and by including a robust matrix of commercially available materials. Unmodified biochar was demonstrated to control the flow of $NH_4^+$ primarily through chemical affinity. Ammonium retention was linked to biochar properties such as high CEC, O/C ratios, ash content, and the presence of oxygen-containing surface functional groups. Nitrate transport was shown to be influenced by physical rather than chemical means. This effect could perhaps be optimized by producing biochars, like SW500, which minimize CEC

but maximize microporosity and surface area, to encourage the physical entrapment of $NO_3^-$. Biochar also had a large effect on saturated hydraulic conductivity, though this effect was not consistent across biochars and soils. Broadly, the results of this study suggest that biochar may increase the residence time of water in sandy soils and increase drainage in fine textured soils, though soil- and biochar- specific investigation is required.

This study demonstrates that biochar can provide a suite of agronomic benefits, from nutrient retention to improvements in soil-water dynamics for crop production. Additional research and quantitative analysis at the micron and sub-micron scale is required to assess the influence of biochar on soil porosity and pore architecture. Field-scale investigation using these soils and biochars is also ongoing, in order to link the impact of biochar on hydraulic conductivity and nutrient leaching to its influence on crop yield and nutrient use efficiency. The findings from this study highlight the need to conduct similar

laboratory, and field, experiments with tailored biochars for specific outcomes, such and nutrient retention, which may show greater efficacy.

**Data and code availability**

All data that supported the results of this publication are publicly available under the Interdisciplinary Earth Data Alliance (IEDA) v1.0 (https://doi.org/10.26022/IEDA/112008). Code is available from the corresponding author upon request.

**Author contribution**

Funding acquisition was carried out by SJP. DLG, SJP, MAN, and DAR conceptualized and designed the experiments. SJP carried out FTIR analysis and data visualization. DAR captured and reconstructed microCT images. JEP conducted PZC analysis. All other experiments were carried out by DLG, under the supervision of SJP and MAN. DLG and IHA conducted formal analysis and data visualization. DLG prepared the manuscript, with equal contributions from all other co-authors.

**Competing interests**

The authors declare they have no conflict of interest.

**Acknowledgements**

We are grateful to Cool Planet, Pacific Biochar, Karr Group Co., and Premier Mushroom for providing the biochars used in this study. We would like to thank Dula Parkinson and Andrew McElrone for their assistance at the Lawrence Berkeley
National Laboratory Advanced Light Source (ALS) Beamline 8.3.2 microtomography facility. We are grateful to Mike Marsh and the staff at Object Research Systems for providing a license to Dragonfly software and for the technical support in reconstructing and interpreting microCT images. Thank you also to the Davis R Users group for statistical and coding consultation, and to Tad Doane and Alex Barbour for laboratory support.

**Financial support**

This publication was made possible by funding from the California Department of Food and Agriculture Fertilizer Research and Education Program (16-0662-SA-0), the Almond Board of California (17-ParikhS-COC-01), and the United States Department of Agriculture (USDA), National Institute of Food and Agriculture (NIFA) through Hatch Formula Funding (CA 2076-H) and multistate regional project (W-3045). ALS is supported by the Director, Office of Science, Office of Basic Energy Science, of the US Department of Energy under contract no. DE-AC02- 05CH11231. Additionally, this research was supported
by the UC Davis Dissertation Year Fellowship, a Henry A. Jastro Graduate Research Award, the Beatrice Oberly and S. Atwood McKeehan Fellowship, and the Foundation for Food and Agriculture Research Fellowship.

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
