# Peer review of "Biochar alters hydraulic conductivity and impacts nutrient leaching in two agricultural soils"

_SOIL, 2021_

## Author Comment (AC1)

Reviewer #1

**The authors contribute with their study to an ongoing and substantial discussion of the effect of biochar on the hydraulic properties of soils and the potential of biochar to bind and retain nitrate and ammonium in soils. While this is an important discussion for the application of biochar in agricultural soils, the submitted manuscript is not well structured and, much more importantly, it is not clearly providing a novel approach or understanding for the ongoing scientific discussion. Furthermore, the manuscript is not transparent to follow the methodological approach. It is not clear why the column retention experiment was only performed for the HSL and the described effect of additional nitrate leaching with biochar is not supported by shown data. The fairly short discussion is by far not complete. Many aspects contradicting the here reported findings are not considered (please see specific comments). This results also in a lack of new mechanistic understanding and the link to the agricultural soils. For example, the authors are not considering the effect of the two agricultural soils on the nutrient mobility or bring their findings in context of potential field applications. I highly recommend the authors to consider a critical discussion of their findings, developing supported mechanistic understanding from these experiments, improve the transparency of the experimental approach and improve the overall manuscript structure. Given these aspects, I decided to reject the current manuscript for publication in SOIL.**

Response: We greatly appreciate the time and effort that went into this extensive and constructive review. We will address the reviewers concerns by restructuring the manuscript, providing more comprehensive information in the materials and methods section, making linkages between these experiments and our corresponding field trials clearer, and highlighting the novelty of this work through a more nuanced, critical, and lengthy discussion section, as detailed below. Furthermore, we have performed an additional experiment to answer the reviewer's questions about the effect of pH and the mechanistic underpinnings behind observed results. The point of zero charge (PZC) for each biochar is now included, as described below.

**Specific comments:**

**Abstract and introduction:**

- **Line 9-10: specify "saturated hydraulic conductivity (Ksat)"**

  Response: We corrected this in the manuscript.

- **Line 44-46: Provide reference for this statement**

  Response: We have included a reference to Clough and Condron (2010) and Peiris et al. (2018), as cited in the bibliography at the end of this response.

- **Line 65: What is the mechanism for the high values found in Yin et al (2018). Please provide more details**

  Response: The mechanism cited in Yin et al. (2018) is the "abundant surface functional groups" that develop at low pyrolysis temperatures. Following our statement on lines 66-74, there is a well-cited review of mechanisms which details the relationship between pyrolysis temperature and biochar characteristics. To make the relationship between this statement and the Yin paper more clear, we made two corrections. On line 69, we added Yin et al. (2018) to the discussion of studies which find higher adsorption values at lower temperatures, to state:

"Lower temperatures have been correlated with higher cation exchange capacity (CEC) (Gai et al., 2014a), higher O/C ratios (Yang et al., 2017), and more abundant surface functional groups (Yin et al., 2018)."

We have also revised the original statement to exclude specific adsorption values, as follows:

"While most reported $Q_{max}$ values are less than 20 mg $NH_4^+$-N $g^{-1}$ (Zhang et al., 2020), higher values have been observed (Yin et al., 2018, Gao et al., 2015)."

This allows for discussion of the range and inconsistencies found within the literature, without overburdening the reader with specific values and mechanisms for each of the 15 cited papers, immediately prior to a 9-line discussion and summary of mechanisms. Furthermore, this revised sentence matches the format provided for the discussion of nitrate sorption on line 55.

- **Line 96-102: Too detailed method description for an introduction. Please shorten to avoid repetition.**

  Response: We have removed 7 lines, to develop the paragraph below:

  "In this project, biochar characterization, sorption, and soil column experiments were carried out using biochars of diverse feedstocks and production temperatures, in order to determine to what degree these biochars: 1) chemically bind nitrate and ammonium; 2) physically alter the soil to influence saturated hydraulic conductivity; or 3) influence nutrient leaching, through either chemical or physical means. This information was used to determine the parameters that may optimize hydrologic and nutrient retention benefits in two agricultural soils, and to investigate the combination of chemical and physical mechanisms by which these benefits are delivered. Our results are expected to inform the process of biochar production or modification for the above-mentioned specific purposes, as well as improve predictions on biochar behaviour in specific agricultural conditions."

**Material and Methods:**
- **Line 106-108: What are the production durations of the selected chars**

  Response: As the biochars are commercially available, many of the production details are proprietary and were not disclosed. However, we can amend the paragraph to state the individual producers, as below. This would allow for other scientists to repeat experiments with these biochars, and to contact the companies for more information if desired. It also emphasizes that individual production details were beyond the control of the authors.

  "Seven biochars were obtained from the following feedstocks and produced at the following temperatures: almond shell at 500 °C (AS500, produced by Karr Group Co.), almond shell and 800 °C (AS800, Premier Mushroom and Community Power Co), coconut shell at 650 °C (CS650, Cool Planet), softwood at 500 °C (SW500, Karr Group Co.), softwood at 650 °C (SW650, Cool Planet), and softwood at 800 °C (SW800, Pacific Biochar), and an additional softwood biochar produced at 500 °C and inoculated with a proprietary microbial formula (SW500-I, Karr Group
Co.).”
- **Line 107-108: Please provide details on the char with inoculated microbial formula**

Response: As stated above, the microbial formula is proprietary and was not disclosed to authors. That fact has now been made clear in the above text. Additionally, as this product is commercially available, the experiment can be reproduced by other researchers by purchasing this material.

- **Line 113: What is the duration of the individual temperature steps?**

The sentence has now been revised to contain the requested information as well as the reference for this method, as below:

“…and moisture, volatile, and ash content were measured as a percent of total dry weight through sequential shifts in furnace temperature (briefly, 2 h at 105 °C, 6 m at 950 °C, and 6 h at 750 °C, respectively) (ASTM D 1762-84, 2011).”

- **Line: 122: The authors can avoid to mention a private company because they followed the standardized protocol/ Line 123: Provide ISO number here**

Response: The company name has been removed, and the ISO number has been included, as follows:

“Specific surface area was determined from $CO_2$ adsorption isotherms according to the Brunauer, Emmet, Teller (BET) method ISO 9277:2010 (International Organization for Standardization (ISO), 2010).”

- **Line 124-128: Not clear if the authors used finally the DRIFT or FTIR. Please clarify.**

Response: DRIFT (diffuse reflectance infrared Fourier transform) is a specific sampling method of FTIR (Fourier transform infrared) spectroscopy; DRIFT was used as the FTIR sampling method. We have edited the FTIR method to be explicitly clear. The text now reads:

“Fourier transform infrared (FTIR) spectra of AS500, AS800, and SW500 biochars were collected using the diffuse reflectance infrared Fourier transform (DRIFT; PIKE Technologies EasiDiff) sampling mode with air dried samples diluted to 3% with potassium bromide.”

- **Line: 137-138: Not clear to what field trials they authors are referring here. Were the soils taken from long-term field trials locations?**

Response: We believe the below paragraph makes clear that the soils were taken from field trials. However, we have modified the sentence (added details in bold) to provide more detail:

“Hanford sandy loam (HSL) and Yolo silt loam (YSiL) soils were chosen for continuity between laboratory experiments and ongoing **3-year** field trials **utilizing the same biochars and soils**. Collectively, these soils represent over 260,000 hectares of arable land in California and offer textural distinctions within a range of soils commonly farmed in the Central Valley of California (Soil Survey Staff, 2014). Soils were located via Web Soil Survey
(http://websoilsurvey.sc.egov.usda.gov/) and collected from the top 30 cm in fallowed
agricultural fields in Parlier, California (HSL) and Davis, California (YSiL)."

• **Line 147: What was the core volume?**

Response: This information has been added to line 176 in Methods section 2.4 *Column experiments*,
as follows below (in bold):

"To investigate the influence of biochar on saturated hydraulic conductivity ($K_{sat}$), constant head
column experiments were performed in five replicates using the 5 station Chameleon Kit
(Soilmoisture Equipment Corporation (SEC) 2816GX). SEC tempe cells**, each with a volume of
136.4 cm$^3$** were packed with soils amended with 0 and 2% (w/w) AS500, AS800, or SW500
biochars…"

• **Line 165: The authors should include more information about the tested models in the
supplement. Which fitting parameter were considered to evaluate the goodness of fit and avoid
over parameterization (e.g. AICc)**

Response: Our simple linear models did not have random effects or interactions, and are therefore
not at danger of being overfit. We have made our statistical approach clearer by rephrasing (new
content in bold):

**"All data were analysed with linear models (lm(response variable ~ biochar))** and one-way
analysis of variance (ANOVA) in the stats and Tidyverse packages in R (R Core Team, 2020;
Wickham et al., 2019). **When more than one soil type was tested (as in $K_{sat}$ measurements),
separate models were built for each soil type to determine the effect of biochar within soil
types.** For analysis of results, all effects with p-values < 0.05 were considered significant. P-
values were generated using the emmeans package in R (Lenth, 2019) and corrected for
multiple comparisons using Tukey's honestly significant difference (HSD) method. Plots were
generated in R using the ggplot2 package (Wickham, 2016) and visualized as the mean plus or
minus the standard error of the means."

• **Line 176-177: Was the soil and biochar homogeneous mixed? How was this ensured?**

Response: Yes, soils and biochars were thoroughly and homogenously mixed through a combination
of stirring and shaking within a sealed container for a minimum of 120 seconds. The following
sentence has been added to line 178:

"Soils and biochars were thoroughly and homogenously mixed prior to being added to tempe
cells."

• **Line 177: What would be the typical application rate on the agricultural soils used in this study?**

Response: The biochar community has yet to reach consensus on recommended application rates,
and therefore there is no typical or standard rate used in agricultural soils. One meta-analysis
concluded that the greatest agronomic benefits were observed in studies utilizing 100 t ha$^{-1}$ (Jeffery
et al., 2011). More recently, Oladele (2019) developed a soil quality index using data from a three- year field trial, which concluded that a biochar application rate of 6-12 t ha$^{-1}$ was optimal, though
results were constrained to acidic alfisols (USDA Soil Taxonomy) (Oladele, 2019). Pandit et al. (2018)
conducted an economic analysis which included payments for C sequestration, to determine an
optimal rate of 15 t ha$^{-1}$ (Pandit et al., 2018). Guo (2020) made even more specific recommendations
based on the results of a literature review and from greenhouse trials, concluding that biochar
should be applied at a concentration of 2–5% by weight for wood- and crop residue-derived
biochars, and 1–3% for manure-derived biochars (Guo, 2020). Our rate of 2% falls within the
recommendations contained within Guo (2020) and Jeffery (2011). If incorporated to a depth of 12
inches, assuming a bulk density of 1.33 g cm$^{-3}$, this is an application rate of 81.2 t ha$^{-1}$. This rate was
chosen as the result of the studies herein mentioned, and as described in the text, is "the midrange
of those represented in similar experiments (Blanco-Canqui, 2017)." The paper cited here is a
literature review that contains data from 28 experiments on biochar's effect on K$_{sat}$.

•   **Line 181: Why did the authors not include both soils here? Please provide a clear argument since**
**this is substantial for the whole discussion of the manuscript.**

We agree that the study would have benefited from leaching data from both the YSiL and the HSL
soils. However, column experiments were performed without the aid of autosamplers or
mechanization of any kind. As shown in Figure 3, the K$_{sat}$ for the unamended HSL soil columns was
1.2 cm s$^{-1}$ . To manually collect 20 pore volumes of leachate, 15 hours of active maintenance was
required per treatment, for a total of 4 treatments. The YSiL K$_{sat}$ was 0.044 cm s$^{-1}$, or a 96%
reduction in flow rate from the HSL. Therefore, we considered it unfeasible for someone to stay in
lab for the time required to collect 20 pore volumes at this speed. After attempting it for two
treatments, we decided to proceed with the HSL data, as it was logistically possible and would
provide more valuable information. To make this clear to readers, we will include the following
statement (new content in bold):

"Columns were also used to investigate the nutrient retention and leaching in HSL
amended with 0 and 2% biochar. Preliminary trials with the YSiL demonstrated that leaching
rates were very low (~0.044 cm$^{-1}$) creating logistical challenges for conducting these
experiments. Additionally, the impact of nitrate leaching is much more pronounced in more
coarsely textured soils and thus leaching experiments were conducted only in HSL columns"

•   **Results: Large parts of the result section describes the biochar and soil. The author should**
**consider to include the characterizations in the material and method section. The result section**
**should focus on the actual findings regarding the sorption and Ksat effect of the biochar on the**
**soils.**

Response: As one of our objectives was to "determine the soil and biochar parameters which may
optimize hydrologic and nutrient retention benefits in two agricultural soils," we do not agree that
this information is extraneous or takes away from the results that follow. This is especially true given
how important IR and microCT data was for interpreting those results. However, we have shortened
this section by moving *Table 2: Functional group assignments corresponding to organic biomass* to
supplementary information.

•   **Line 196: Please specify "carbon, hydrogen contents and leachable DOC"**

Response: We are not sure why this is different than the phrase already included ("decreased
carbon, hydrogen, and DOC") as the word "contents" is implied when discussing biochar
constituents, and "leachable" is both implied from the OC having been dissolved (D), and explicit
from the description of DOC methodology.
•  **Line 196-197: Please avoid interpretation of the data and comparison to the litterateur in the**
**result section. This is part of the discussion.**
Response: We have addressed this in the manuscript by moving any interpretation and comparison
to the literature to the discussion section. This will further shorten the biochar characterization
results by two lines.
•  **Line 203-204: This aspect should be considered in the discussion and clearly mentioned in the**
**material and methods. The oxidation state of the biochar will also influence the surface reactivity,**
**which may, in fact, explain the here observed findings.**
Response: We agree that the oxygen content (interpreted from "oxidation state" in reviewer
content) of the biochar will influence surface reactivity and explain results. As such, we covered this
extensively in the discussion section beginning on line 317:
While it is typical for biochars produced at high temperatures to have low O/C ratios and low
CEC (Hassan et al., 2020), AS800 had the largest O/C ratio at 0.56 (presumably due to post-
pyrolysis oxidation), and the second highest CEC at 52.75 cmolc kg$^{-1}$. These properties, as well as
the IR band at 1405 cm$^{-1}$ (COO$^{-}$), likely explain the high ammonium retention, as they indicate
increased exchange sites and oxygen-containing functional groups which can react with
ammonium. The relationship between these biochar properties and ammonium binding capacity
was also demonstrated with SW800, which had the highest CEC at 60.83 cmolc kg$^{-1}$, the second
highest O/C ratio at 0.27, and the second highest ammonium binding capacity. These
observations are consistent with those of other studies (Gai et al., 2014; Yang et al., 2017).
As the specific details of biochar production methodology were proprietary, this is not a "method"
but a hypothesis to explain an observed result. As such, it cannot be included in the methods
section. However, we have made this more clear by removing the line about post-pyrolysis oxidation
from the results section and including it strictly in the discussion section.
•  **Line 211: include "1410 and 1418 cm$^{-1}$"**
Response: We will correct this by adding "cm$^{-1}$" in the manuscript, and thank the reviewer for this
attention to detail.
•  **Line 213-215: As mentioned above these differences in biochar production should be clearly**
**presented in the material and method section and also critically discussed in the discussion**
Response: We agree, and will move this section into the materials and methods.
•  **Line 237: Soil texture expressed as mass per mass (g/g) is a content and not a concentration.**
**Furthermore, avoid digits for these values.**

We agree that percent soil texture should be reported as content and have made this change.
• **Table 3: Correct the number of digits for texture. Also, pH is commonly measured with on digit**
**precision.**
Response: We have made these changes as recommended.
• **Section 3.3: Provide the data for the nitrate leaching. What is the order of magnitude if the nitrate**
**release? This data needs to be shown.**
Response: This data has been added to the supplementary information document.
• **Figure 2: Please show the fitted isotherms**
Response: We initially visualized Freundlich and Langmuir models for each biochar in figure 2.
However, due to the high number of biochars included in this study, the figure became cluttered
and difficult to read. While we acknowledge that fitted isotherms are one appropriate way to display
sorption data, there is a rich literature base which shows Ce vs Qe, or % adsorbed vs quantity in
solution, without model fits, but rather provides $R^2$ values for models instead (Gai et al., 2014; Wang
et al., 2015; Yao et al., 2012, to name just a few). Due to the relatively low $R^2$ values we discuss, we
believe simply visualizing Ce vs Qe for each rep of each treatment is a more descriptive and
quantitative way of viewing this data, with model $R^2$ values in a table provided directly following the
figure.
• **Line 268-369: Please specify this statement and clearly indicate to which the p values correspond**
**to.**
Response: We have edited the statement as follows:
"There was a main effect of biochar (p = 0.001) and soil texture (p < 0.001), as well as a
significant interaction between biochar and soil texture (p = 0.006), on saturated hydraulic
conductivity."
• **Line: 285: "HSL at pore volume 14.3" corresponds this to the controls?**
Response: We have edited the statement to say: "HSL (control) at pore volume 14.3"
• **Discussion: This discussion is not complete and is not discussion available contradicting literature.**
**A few suggestions can be found below. However, I recommend an extensive literature review to**
**develop a structured and complete discussion.**
We agree that the manuscript could be improved by a lengthier, more nuanced, and more detailed
discussion, and thank the reviewer for this suggestion. However, we believe that this manuscript
already includes an extensive literature review, covering most of the articles the reviewer
suggested. We do not believe we need *more* literature, but, as the reviewer stated, a *better*
*structured* literature review. Currently, we have included a lengthy discussion of the contradictory
literature in the introduction. We did not include these same references in the discussion so as to avoid repetition, but agree that this context is important for our specific results. In the revised
version, we have moved some of the extensive discussion from the introduction into the discussion,
and relate all findings to our results, as described extensively throughout this document.

- **Line 315: t is mentioned already that this char might be oxidized, the authors should clearly indicate this in the sections before. This initial "bias" effect needs more critical discussion here.**

Response: We have revised the information about potential post-pyrolysis oxidation as described in this document on page 5, regarding the comment about line 21.

- **The hole paragraph provides no mechanist discussion. It is just comparing the findings with the literature. Please improve the discussion here and connect the different sportive capacities with the properties of of the chars.**

Response: We respectfully disagree that this paragraph does not include mechanistic discussion, as we clearly delineate the relationship between biochar properties (high O/C, CEC, and oxygen-containing function groups) and their demonstrated ability to retain positively charged ammonium ions, as copy/pasted below:

> "AS800 had the largest O/C ratio at 0.56 (presumably due to post-pyrolysis oxidation), and the second highest CEC at 52.75 cmolc kg$^{-1}$. These properties, as well as the IR band at 1405 cm$^{-1}$ (COO$^-$), likely explain the high ammonium retention, as they indicate increased exchange sites and oxygen-containing functional groups which can react with ammonium. The relationship between these biochar properties and ammonium binding capacity was also demonstrated with SW800, which had the highest CEC at 60.83 cmolc kg$^{-1}$, the second highest O/C ratio at 0.27, and the second highest ammonium binding capacity. These observations are consistent with those of other studies (Gai et al., 2014; Yang et al., 2017). No clear trends between surface area and ammonium retention emerged in this study."

- **Line 318-320: Figure 2 shows actually no clear differences between SW800 and the other chars. What is the explanation? In fact, only AS800 shows the previous mentioned large binding capacities of ammonium.**

Response: Figure 2 visibly demonstrates that SW800 has a higher binding capacity than all biochars (except AS800) at initial ammonium concentrations of 50, 100, and 200 mg L$^{-1}$. The reviewer is correct in their statement that this effect is less clear at the higher concentrations of 400 and 600 mg L$^{-1}$. Furthermore, authors of this manuscript never claimed any biochar to have a large binding capacity, but rather stated: "all biochars exhibited the capacity to remove ammonium from solution (Fig. 2), though K$_f$ values were low (Table 4)" and "The ability of all seven biochars to retain ammonium, and within the demonstrated ranges, is consistent with other published studies (Zhang et al., 2020). AS800 exhibited substantially higher ammonium binding capacity than the other biochars tested." These statements are in agreement with those the reviewer made in this comment.

- **Line 324-329: The whole paragraph misses to bring the findings of this study in context of studies with contradicting results which is actually in some of the already cited papers. But there is certainly more literature on this effects and higher nitrate binding capacities are reported. Only Zhang et al (2020) is cited here to support the findings of this study, which is by far not complete. Here are a few suggestions also providing contradictory findings (and literature within):**

•   *Kameyama, K., Miyamoto, T., Iwata, Y., and Shiono, T.: Influences of feedstock and pyrolysis*
*temperature on the nitrate adsorption of biochar, Soil Science and Plant Nutrition, 62, 180–184,*
*https://doi.org/10.1080/00380768.2015.1136553, 2016.*
•   *Cao, H., Ning, L., Xun, M., Feng, F., Li, P., Yue, S., Song, J., Zhang, W., and Yang, H.: Biochar can*
*increase nitrogen use efficiency of Malus hupehensis by modulating nitrate reduction of soil and*
*root, Applied Soil Ecology, 135, 25–32, https://doi.org/10.1016/j.apsoil.2018.11.002, 2019.*
•   *Yang, J., Li, H., Zhang, D., Wu, M., and Pan, B.: Limited role of biochars in nitrogen fixation through*
*nitrate adsorption, Science of The Total Environment, 592, 758–765,*
*https://doi.org/10.1016/j.scitotenv.2016.10.182, 2017.*
•   *Aghoghovwia, M. P., Hardie, A. G., and Rozanov, A. B.: Characterisation, adsorption and desorption*
*of ammonium and nitrate of biochar derived from different feedstocks, Environmental Technology,*
*1–14, https://doi.org/10.1080/09593330.2020.1804466, 2020.*
•   *Hagemann, N., Kammann, C. I., Schmidt, H.-P., Kappler, A., and Behrens, S.: Nitrate capture and slow*
*release in biochar amended compost and soil, PLoS ONE, 12, e0171214,*
*https://doi.org/10.1371/journal.pone.0171214, 2017.*

Response: We agree that there are many studies which report results contradictory to our own, and as
the reviewer mentioned, many were cited in this paper. Zhang et al (2020) was cited as the only source
for the statement on line 324 because, as explicitly stated, this study is a literature review which
calculated mean nitrate sorption for a range of biochars across the literature. However, we have revised
this statement to include more of the sources already cited, as well as those the reviewer has offered, to
make our knowledge of the literature base more explicit. Furthermore, our introduction section
currently includes 11 sources to support the discussion of contradictory nitrate sorption. To present a
clearer argument and provide a better manuscript structure, we will reorganize the material aiming for a
shorter introduction and a lengthier discussion section, in which each of our results are directly linked
with the studies that found similar or contradictory sorption.

•  **Section 4.2: Similar to the paragraph before, this section misses a critical discussion of the findings.**
**The authors need to include a more mechanistic explanation of the ammonium and nitrate retention**
**in soils. Actually, the soil effect (e.g. texture and pH) is not included at all. All these observations are**
**also based on the experiment of the HSL. This need to be critically discussed. The effect may change**
**drastically with different soils. Please follow also here the above mentioned literature, which is only a**
**short list of literature on this topic.**

Response: As stated in the previous response, we have reorganized the manuscript to include the
extensive discussion that is currently in place on lines 45-74 of the introduction section, in the discussion
section instead. This discussion includes detailed descriptions of mechanisms from contradictory results
in the literature:

"Due to the deprotonation of surface functional groups at agronomic soil pHs, biochar is typically
negatively charged….. Electrostatic repulsion between nitrate and biochar has indeed been regularly
cited as the reason behind little to no nitrate removal in batch sorption experiments… Higher $Q_{max}$
values for biochar and ammonium are to be expected, as ammonium exists in the cationic form in
aqueous environments and would more readily adsorb to negatively charged biochar surfaces…..
Multiple authors have observed that sorption capacity decreases with increasing production
temperature (Gai et al., 2014; Gao et al., 2015; Yin et al., 2018). Lower temperatures have been
correlated with higher cation exchange capacity (CEC) (Gai et al., 2014), and higher O/C ratios (Yang et al., 2017). These properties may contribute to biochars with the ability to remove ammonium
from solution, as they provide a greater number of exchange sites and oxygen-containing functional
groups which can react with ammonium (Yang et al., 2017). The reverse trend has also been
observed, however, with authors noting that an increase in production temperature resulted in
higher ammonium $Q_{max}$ values (Chandra et al., 2020; Zeng et al., 2013; Zheng et al., 2013). These
authors point towards the higher specific surface area (SA) of biochar at higher production
temperatures as a critical parameter to predicting ammonium adsorption."

To further address the effect of soils not tested in this experiment, we have included a critical discussion
of the impact of soil texture as demonstrated in other experiments, and explicitly state that our results
are constrained to a sandy loam, and may not be observed in other contexts. To address the pH effect of
various soils, we have conducted one additional experiment to learn the point of zero charge (PZC) of
the three biochars in question. We have included this data, as well as the appropriate methods
description and citation of sources. Briefly, we found that the PZC was 6.8 for AS800, 3.2 for AS500, and
3.9 for SW500. As most agricultural soils have a pH well above 4, the behavior of AS500 and SW500 are
not likely to change as the result of agricultural soil pH, as thereactive functional groups on soil organic
matter and minerals will remain deprotonated and able to bind to ammonium more strongly than
nitrate. The higher PZC of AS800 was to be expected, as it has a higher ash content, and higher metal-
oxide content as demonstrated through IR peaks at ~1000 to 700 cm$^{-1}$, consistent with metal oxide
vibrations (Parikh et al., 2014). That the pH of AS800 is closer to the soil pH of those tested in this study
(7.3), however, indicates that AS800 may be strongly effected by soil pH, and able to bind even more
ammonium at lower pHs. We will expand our mechanistic discussion to include this information and
citation of the effect of soil pH on the electrostatic affinity between biochar and nitrate and ammonium.

• **The authors also miss to bring their findings in context of the applicability under field conditions and**
**unsaturated soil conditions.**

Response: We agree that the link between this study and our ongoing field trials was not made clear
enough, as description of the field trials is currently contained only in the methods section 2.2. In the
next iteration of this manuscript, we have included an additional final paragraph in the discussion
section, as detailed below:

4.4 Implications for field conditions

The results of this study suggest these biochars may increase the residence time of water in
sandy soils and increase drainage in fine textured soils during irrigation or flooding events, or
when soils are otherwise saturated. Results may be particularly relevant for flooded agricultural
systems such as rice, where ammonium is the primary source of N and water retention is a key
parameter for success (Minami, 1995). Indeed, 95% of California rice production occurs in the
Sacramento Valley, where both the YSiL and HSL soils are common
(http://rice.ucanr.edu/About_California_Rice/). Data from these trials may help growers in
these regions and soil textures determine if biochar can increase water and nutrient retention in
their systems.

Recent meta-analyses have concluded that biochar substantially increased soil water content at
field capacity and permanent wilting point, in the field and lab, in coarse textured soils only
(Blanco-Canqui, 2017; Razzaghi et al., 2020). Despite these observed trends, benefits have also
been observed in fine textured soils, including reduced crop water stress, increased yield (Kerré

et al., 2017; Nawaz et al., 2019), and reduced crop loss during deficit irrigation (Madari et al.,
2017). Other authors have reported little to no effect, or transient effects, of biochar on soil
water dynamics in both fine and coarse textured soils (Jones et al., 2012; McDonald et al., 2019;
Nelissen et al., 2015). However, results from our experiments cannot be extrapolated to dryland
agriculture or in soils that experience wet-dry cycles, as unsaturated hydraulic conductivity was
not measured. In order to determine how these biochars may behave in unsaturated conditions,
three-year processing tomato field trials are currently underway in these same soil textures, in
which soil-water dynamics are being measured.

• **Section 4.3: This section also misses some aspects which need to be discussed in this context. Only**
**one application rate of biochar was used, it is not discussed if this rate is representative for these soils**
**and its acricultural use. Furthermore, it is known that also the application rate and particle size has an**
**effect on the Ksat depending in the soil texture as discussed in the below listed literature.**
• *Obia, A., Mulder, J., Hale, S. E., Nurida, N. L., and Cornelissen, G.: The potential of biochar in*
*improving drainage, aeration and maize yields in heavy clay soils, PLoS ONE, 13, e0196794,*
*https://doi.org/10.1371/journal.pone.0196794, 2018.*
• *Herath, H. M. S. K., Camps-Arbestain, M., and Hedley, M.: Effect of biochar on soil physical properties*
*in two contrasting soils: An Alfisol and an Andisol, 209–210, 188–197,*
*https://doi.org/10.1016/j.geoderma.2013.06.016, 2013.*
• *Barnes, R. T., Gallagher, M. E., Masiello, C. A., Liu, Z., and Dugan, B.: Biochar-induced changes in soil*
*hydraulic conductivity and dissolved nutrient fluxes constrained by laboratory experiments, 9,*
*https://doi.org/10.1371/journal.pone.0108340, 2014.*

Response: We agree that application rate and particle size are important determinants of nutrient
retention and hydraulic conductivity in biochar-amended soils, and will include these and other citations
in a brief discussion of this. However, as described on page 4 of this document (in response to the
comment about line 177), there is no current "representative" biochar amendment rates for particular
uses or soil types. The chosen rate is representative of recommendations that exist in the literature (see
page 4), and is the midrange from experiments of similar design (See tables in literature review from
Blanco-Canqui, 2017). This study measured several responses ($K_{sat}$ in two soils, nitrate and ammonium
leaching (quantity and timing) in one soil, and nitrate and ammonium sorption, using 7 biochars in which
we tested the effect of feedstock and production temperature). The effect of application rate was
outside the purview of this study, given the extensive work already involved in the experimental design.
Furthermore, we did not test the effect of particle size by creating biochars of different sizes, because
we sought to use commercially available materials so that experiments could be repeated. This is, in
part, in response to a literature review which critiqued biochar studies which use only small-batch lab-
created biochars (Zhang et al., 2016). Nevertheless, we included a discussion of particle size in lines 353-
359 when describing hydraulic conductivity. As stated previously, we will lengthen the discussion around
these topics by moving citations from the introduction and by making the link between our results and
current literature more explicit.

• **Line 353-354: What was the relative particle size distribution. These characteristics are not presented.**

Response: Mean and median particle sizes for all biochars are provided in Table 1.

• **Line 354-355: How can the authors provide prove of this statement?**

This statement is a hypothesis backed by evidence from the literature, but cannot be proved within the
context of our study. As stated, this statement could be further explored and supported through future
research: "Additional research and quantitative analysis at the micron and sub-micron scale is required
to assess the influence of biochar on soil porosity and pore architecture."

526•   **Line 374-376: This has not been discussed so far. But the field applications of this experiment need to**
**be included in the critical discussion. The intention of this study was, according to the title, to consider**
**agricultural soils. Furthermore, how can the authors draw a conclusion for flooded agricultural**
**systems when they did not include soils from such systems?**

Response: As described on line 435 of this document (in response to section 4.3), we will add another
section to the discussion entitled "4.4 Implications for field conditions".

**Summary**
We again thank the reviewer for these detailed and helpful comments, which we believe will strengthen
the manuscript, broaden its impact, and increase interest from readers of SOIL. To address the
reviewer's primary concerns, we have restructured the discussion which was previously split between
the introduction and discussion sections, clarified many details of the materials and methods, and better
linked these experiments to production-scale agriculture.

Though the reviewer critiqued the lack of discussion and mechanistic investigation, we believe the error
is not in a *lack* but in a *non-ideal placement.* We have moved the already cited sources and descriptions
from the introduction, and better connected them to our own results in the discussion section. As the
reviewer described, the current structure of the manuscript is not as strong as it could be. We have
rearranged according to the reviewer's suggestions as described extensively above. Furthermore, we
added data from our additional experiment on PZC, literature sources the reviewer provided as well as
others not provided, and better connected these results to our ongoing and critical field trials. While we
appreciate the reviewer's suggestions, we respectfully do not believe the comments provided are
grounds for rejection, as there are no issues with experimental design, results, or importance of the
work pursued. We believe we can swiftly implement the provided suggestions for a better structured
and more transparent manuscript, that will be of great impact.

**References**

ASTM D 1762-84: Standard Test Method for Chemical Analysis of Wood Charcoal., ASTM Int.,
84(Reapproved 2007), 1–2, doi:10.1520/D1762-84R07.2, 2011.

Blanco-Canqui, H.: Biochar and Soil Physical Properties, Soil Sci. Soc. Am. J., 81(4), 687–711,
doi:10.2136/sssaj2017.01.0017, 2017.

Chandra, S., Medha, I. and Bhattacharya, J.: Potassium-iron rice straw biochar composite for sorption of
nitrate, phosphate, and ammonium ions in soil for timely and controlled release, Sci. Total Environ., 712,
136337, doi:10.1016/j.scitotenv.2019.136337, 2020.

Clough, T. J. and Condron, L. M.: Biochar and the Nitrogen Cycle: Introduction, J. Environ. Qual., 39(4),
1218, doi:10.2134/jeq2010.0204, 2010.

Gai, X., Wang, H., Liu, J., Zhai, L., Liu, S., Ren, T. and Liu, H.: Effects of feedstock and pyrolysis
temperature on biochar adsorption of ammonium and nitrate, PLoS One, 9(12), 1–19,
doi:10.1371/journal.pone.0113888, 2014.

Gao, F., Xue, Y., Deng, P., Cheng, X. and Yang, K.: Removal of aqueous ammonium by biochars derived
from agricultural residuals at different pyrolysis temperatures, Chem. Speciat. Bioavailab., 27(2), 92–97,
doi:10.1080/09542299.2015.1087162, 2015.

Guo, M.: The 3R Principles for Applying Biochar to Improve Soil Health, Soil Syst., 4(1), 9,
doi:10.3390/soilsystems4010009, 2020.

Hassan, M., Liu, Y., Naidu, R., Parikh, S. J., Du, J., Qi, F. and Willett, I. R.: Influences of feedstock sources
and pyrolysis temperature on the properties of biochar and functionality as adsorbents: A meta-analysis,
Sci. Total Environ., 744, 140714, doi:10.1016/j.scitotenv.2020.140714, 2020.

International Organization for Standardization (ISO): Determination of the specific surface area of solids
by gas adsorption - BET method (ISO 9277:2010(E)), Ref. number ISO, doi:10.1007/s11367-011-0297-3,
2010.

Jeffery, S., Verheijen, F. G. A., van der Velde, M. and Bastos, A. C.: A quantitative review of the effects of
biochar application to soils on crop productivity using meta-analysis, Agric. Ecosyst. Environ., 144(1),
175–187, doi:10.1016/j.agee.2011.08.015, 2011.

Jones, D. L., Rousk, J., Edwards-Jones, G., DeLuca, T. H. and Murphy, D. V.: Biochar-mediated changes in
soil quality and plant growth in a three year field trial, Soil Biol. Biochem., 45, 113–124,
doi:10.1016/j.soilbio.2011.10.012, 2012.

Kerré, B., Willaert, B., Cornelis, Y. and Smolders, E.: Long-term presence of charcoal increases maize
yield in Belgium due to increased soil water availability, Eur. J. Agron., 91(September), 10–15,
doi:10.1016/j.eja.2017.09.003, 2017.

Lenth, R.: Emmeans: estimated marginal means, Aka Least-Squares Means., https://cran.r-
project.org/package=emmeans, doi:https://CRAN.R-project.org/package=emmeans, 2019.

Madari, B. E., Silva, M. A. S., Carvalho, M. T. M., Maia, A. H. N., Petter, F. A., Santos, J. L. S., Tsai, S. M.,
Leal, W. G. O. and Zeviani, W. M.: Properties of a sandy clay loam Haplic Ferralsol and soybean grain
yield in a five-year field trial as affected by biochar amendment, Geoderma, 305(June 2016), 100–112,
doi:10.1016/j.geoderma.2017.05.029, 2017.

McDonald, M. R., Bakker, C. and Motior, M. R.: Evaluation of wood biochar and compost soil
amendment on cabbage yield and quality, Can. J. Plant Sci., 99(5), 624–638, doi:10.1139/cjps-2018-
0122, 2019.

Minami, K.: The effect of nitrogen fertilizer use and other practices on methane emission from flooded
rice, Fertil. Res., 40(1), 71–84, doi:10.1007/BF00749864, 1995.

Nawaz, H., Hussain, N., … M. A.-I. and 2019, U.: Biochar Application Improves the Wheat Productivity
under Different Irrigation Water-Regimes, Intl. J. Agric. Biol, 21(April), 936–942,
doi:10.17957/IJAB/15.0978, 2019.

Nelissen, V., Ruysschaert, G., Manka'Abusi, D., D'Hose, T., De Beuf, K., Al-Barri, B., Cornelis, W. and
Boeckx, P.: Impact of a woody biochar on properties of a sandy loam soil and spring barley during a two-
year field experiment, Eur. J. Agron., 62, 65–78, doi:10.1016/j.eja.2014.09.006, 2015.

Oladele, S. O.: Changes in physicochemical properties and quality index of an Alfisol after three years of
rice husk biochar amendment in rainfed rice – Maize cropping sequence, Geoderma, 353(June), 359–
371, doi:10.1016/j.geoderma.2019.06.038, 2019.

Pandit, N. R., Mulder, J., Hale, S. E., Zimmerman, A. R., Pandit, B. H. and Cornelissen, G.: Multi-year
double cropping biochar field trials in Nepal: Finding the optimal biochar dose through agronomic trials
and cost-benefit analysis, Sci. Total Environ., 637–638, 1333–1341, doi:10.1016/j.scitotenv.2018.05.107,
2018.

Parikh, S. J., Goyne, K. W., Margenot, A. J., Mukome, F. N. D. and Calderón, F. J.: Soil chemical insights
provided through vibrational spectroscopy., 2014.

Peiris, C., Gunatilake, S. R., Wewalwela, J. J. and Vithanage, M.: Biochar for sustainable agriculture:
Nutrient dynamics, soil enzymes, and crop growth, Elsevier Inc., 2018.

R Core Team: R: A language and environment for statistical computing, [online] Available from:
https://www.r-project.org/, 2020.

Razzaghi, F., Obour, P. B. and Arthur, E.: Does biochar improve soil water retention? A systematic review
and meta-analysis, Geoderma, 361(September), 114055, doi:10.1016/j.geoderma.2019.114055, 2020.

Soil Survey Staff: Web Soil Survey, Natural Resources Conservation Service, United States Department of
Agriculture, Nat. Resour. Conserv. Serv. United States Dep. Agric., 1–2 [online] Available from:
http://websoilsurvey.nrcs.usda.gov/ (Accessed 1 January 2021), 2014.

Wang, B., Lehmann, J., Hanley, K., Hestrin, R. and Enders, A.: Adsorption and desorption of ammonium
by maple wood biochar as a function of oxidation and pH, Chemosphere, 138, 120–126,
doi:10.1016/j.chemosphere.2015.05.062, 2015.

Wickham, H.: ggplot2: Elegant Graphics for Data Analysis, Springer-Verlag New York [online] Available
from: https://ggplot2.tidyverse.org, 2016.

Wickham, H., Averick, M., Bryan, J., Chang, W., McGowan, L., François, R., Grolemund, G., Hayes, A.,
Henry, L., Hester, J., Kuhn, M., Pedersen, T., Miller, E., Bache, S., Müller, K., Ooms, J., Robinson, D.,
Seidel, D., Spinu, V., Takahashi, K., Vaughan, D., Wilke, C., Woo, K. and Yutani, H.: Welcome to the
Tidyverse, J. Open Source Softw., 4(43), 1686, doi:10.21105/joss.01686, 2019.

Yang, H. I., Lou, K., Rajapaksha, A. U., Ok, Y. S., Anyia, A. O. and Chang, S. X.: Adsorption of ammonium in
aqueous solutions by pine sawdust and wheat straw biochars, Environ. Sci. Pollut. Res., 25(26), 25638–
25647, doi:10.1007/s11356-017-8551-2, 2017.

Yao, Y., Gao, B., Zhang, M., Inyang, M. and Zimmerman, A. R.: Effect of biochar amendment on sorption
and leaching of nitrate, ammonium, and phosphate in a sandy soil, Chemosphere, 89(11), 1467–1471,
doi:10.1016/j.chemosphere.2012.06.002, 2012.

Yin, Q., Zhang, B., Wang, R. and Zhao, Z.: Phosphate and ammonium adsorption of sesame straw biochars produced at different pyrolysis temperatures, Environ. Sci. Pollut. Res., 25(5), 4320–4329,
doi:10.1007/s11356-017-0778-4, 2018.

Zeng, Z., Zhang, S., Li, T., Zhao, F., He, Z., Zhao, H., Yang, X., Wang, H., Zhao, J. and Rafiq, M. T.: Sorption
of ammonium and phosphate from aqueous solution by biochar derived from phytoremediation plants,
J. ZHEJIANG Univ. B, 14(12), 1152–1161, doi:10.1631/jzus.B1300102, 2013.

Zhang, D., Yan, M., Niu, Y., Liu, X., van Zwieten, L., Chen, D., Bian, R., Cheng, K., Li, L., Joseph, S., Zheng,
J., Zhang, X., Zheng, J., Crowley, D., Filley, T. R. and Pan, G.: Is current biochar research addressing global
soil constraints for sustainable agriculture?, Agric. Ecosyst. Environ., 226, 25–32,
doi:10.1016/j.agee.2016.04.010, 2016.

Zhang, M., Song, G., Gelardi, D. L., Huang, L., Khan, E., Parikh, S. J., Ok, Y. S., Song, G., Gelardi, D. L.,
Huang, L., Khan, E., Parikh, S. J. and Ok, Y. S.: Evaluating biochar and its modifications for the removal of
ammonium, nitrate, and phosphate in water, Water Res., Preprint, doi:10.1016/j.watres.2020.116303,
2020.

Zheng, H., Wang, Z., Deng, X., Zhao, J., Luo, Y., Novak, J., Herbert, S. and Xing, B.: Characteristics and
nutrient values of biochars produced from giant reed at different temperatures, Bioresour. Technol.,
130, 463–471, doi:10.1016/j.biortech.2012.12.044, 2013.

---

## Author Comment (AC2)

Response: We thank the reviewer for their thoughtful and in-depth analysis of our manuscript, and appreciate the time and effort that went into this review. We have addressed the reviewer's comments by restructuring the manuscript to shorten the introduction and lengthen the discussion. Those changes make our knowledge of the literature more explicit, and better situate our specific results within the ongoing work in this field. We have also made significant edits to the materials and methods section which clarify many details that were previously opaque. Finally, we have performed an additional experiment at the reviewer's suggestion, to include point of zero charge (PZC) data for the biochars included in the column studies. As there was overlap in the suggestions from each reviewer, some of our responses can also be found in the document submitted in response to reviewer #1. Details of our edits are included below.

**Application of biochar to bind nutrients in soil and alter hydraulic properties of the soil is an important and relevant topic for large scale application of biochar in agricultural fields. The authors of this current paper have tried to add more insights into the existing literature in this context. Overall, after a first glance through, the reader can follow the main message of the paper. However, I have a few main points of concern regarding the manuscript:**

- **The title of the paper states "inhibits nutrient leaching" – the data for nitrate does not necessarily show this.**

  Response: We agree that ammonium is retained to a much larger extent, and has greater potential for leaching mitigation in biochar-amended soils. However, nitrate leaching was also inhibited, though to a lesser and more transient extent. While this result is minor in magnitude, it may have great significance for fertilizer use efficiency in cropping systems, especially where biochars can be engineered to have high surface area and low CEC, as described in this study. To address the reviewer's concerns, and more accurately depict the work contained within this manuscript, we have changed the title to "Biochar alters hydraulic conductivity and **impacts** nutrient retention in two agricultural soils."

- **There lacks a sense of novelty in the experimental approach of the manuscript. Experimental details are missing especially for the column studies.**

  Response: Details have been extensively updated, as described below in response to reviewer's specific comments. Regarding the novelty of this work, we agree that we have not made this clear enough in the original manuscript. In the next iteration, we have explicitly stated that this study is novel for the following reasons:
  1) It includes a robust experimental matrix with 7 commercially available biochars included in the sorption experiments (and, based on those results, 3 biochars in 2 soils for $K_{sat}$ data, and then 3 biochars in a sandy soil for leaching data, where results are most important given the potential for sandy soils to leach N). As stated many times throughout the literature, the use of commercially available materials (as opposed to laboratory-produced biochar) is essential for replicability of results, and for the potential for these materials to be used in real world cropping systems;
  2) The experimental approach allows for chemical and physical retention mechanisms to be distinguished: Even where biochar displayed no chemical affinity for nitrate, nitrate was retained in leaching studies, where it was linked to high surface area and low CEC. This suggests a physical entrapment, as elucidated in-text with appropriate citations. While many studies investigate nitrate/ammonium chemisorption or leaching, rarely are both explored given the extensive labor involved in experimental setup and maintenance. Data from our study will be critical for biochar producers to design materials that improve soil water or nutrient retention dynamics, or for land managers to predict how biochars may behave in specific agricultural conditions.

3) This study utilizes the same biochars and soils as those in 3-year field trials. Results from these lab scale experiments can be used to interpret those obtained from field trials, and help provide both fine resolution mechanistic investigation, and effects from real-world agricultural systems.

**A more mechanistic insight would have been interesting. Key factors which would have been critical for achieving this and making a more impactful statement are (i) measurement of point of zero charge (for supporting any statements using electrostatic repulsion or attraction) (ii) measurements of anions and cations released during column nutrient leaching tests (iii) use of non-reactive tracers such as "deuterated water" could have been an interesting approach to understand movement of water through columns, etc.**

- **The term "physical and chemical interactions/affinity" is used very lightly and often in the manuscript without providing concrete proof for these interactions.**
- **An in-depth literature study in the discussion would have provided readers with more confidence in the conclusions that the authors wished to make.**
- **The entire sense of "timing of release of nitrate" and its importance needs to have been brought to light. Is it sufficient for nitrate to be captured physically for a short duration and then released?**
- **The entire discussion in Section 4.3 is underwhelming.**

Response: At the reviewer's suggestion we have included PZC measurements, and extended the discussion of likely and potential mechanisms by including a more extensive literature review in the discussion, which was previously confined to the introduction. We thank the reviewer for their comments, as they have led to a better structured manuscript which will be of greater impact and interest to readers of SOIL. We have addressed the remaining issues in detail in response to specific comments below.

**Abstract and Introduction**

- **Line 47-48 – this is not always true. There are some biochars which have a PZC of 7.5 or higher and then they might be positively charged.**

Response: We agree that biochars are not always negatively charged, and have changed the statement to include more extensive discussion and citations, as below:

"Biochar surfaces range in their protonation state when added to the soil, as a function of soil pH and their point of zero charge (PZC). While PZCs between 7 and 10 have been observed (Lu et al., 2013; Uchimiya et al., 2011), the high number of oxygen-containing (primarily carboxyl) functional groups typically lead to PZCs between 1.5 and 5 (Peiris et al., 2019; Uchimiya et al., 2011; Wang et al., 2020). Due to these PZCs, the
deprotonation of biochar surface functional groups occurs, leading to a net negative
charge within most agronomic soils (pH ~5-7.5)."

Furthermore, we conducted an additional experiment at the reviewer's suggestion to measure
the point of zero charge (PZC) of the three biochars used in our column studies. We have
included this data, as well as the appropriate methods description and citation of sources.
Briefly, we found that the PZC was 6.8 for AS800, 3.2 for AS500, and 3.9 for SW500. As most
agricultural soils have a pH well above 4, including those tested in our study, AS500 and SW500
would be expected to be negatively charged. The higher PZC of AS800 was to be expected, as it
has a higher ash content, and higher metal-oxide content as demonstrated through IR peaks at
~1000 to 700 cm$^{-1}$, consistent with metal oxide vibrations (Parikh et al., 2014). This PZC is lower
than the pH of the soil it was added to, and was likely negatively charged.

• **Line 85 – Suggestion is to introduce what is saturated hydraulic conductivity out here itself.**

Response: The line currently reads: "biochar has largely been shown to decrease the ability of a
saturated soil to transmit water (saturated hydraulic conductivity ($K_{sat}$))." This statement both
introduces the definition of hydraulic conductivity and the abbreviation it is referred to
throughout the rest of the manuscript. If the reviewer is suggesting something different, it is
unfortunately not clear to us.

• **Line 91 – how is the biochar "physically altering the soil to influence Ksat?"**

Response: Lines 80-89 directly prior to this statement provide a detailed discussion of how
biochar can physically alter soil structure through decreased bulk density, increased porosity,
and changes in mean pore size, and therefore influence water movement through the soil
(below). However, to make our meaning more clear, we will delete the word "physically" from
the sentence on line 91.

"In addition to chemical and microbial mechanisms, biochar may retain N through
physical means (Clough and Condron, 2010). One study determined that biochar
decreased soil bulk density by 3 to 31%, and increased porosity by 14 to 64% (Blanco-
Canqui, 2017). Biochar can also alter mean pore size and pore architecture, thereby
influencing tortuosity and the residence time of water and nutrients within the soil
profile (Lim et al., 2016; Quin et al., 2014). The impact of biochar on hydraulic
conductivity largely appears dependent on soil texture, which highly influences pore
structure. While exceptions have been observed, biochar has largely been shown to
decrease the ability of a saturated soil to transmit water (saturated hydraulic
conductivity ($K_{sat}$)) in coarse textured soils and increase $K_{sat}$ in finer soils (Blanco-Canqui,
2017). The impact of biochar on these soil physical properties may influence nitrate
retention through a mechanism known as "nitrate capture," in which nitrate molecules
become physically entrapped within biochar pores (Haider et al., 2016), potentially
leading to increased residence time in crop rooting zones and a greater opportunity for
plant uptake (Haider et al., 2020; Kameyama et al., 2012; Kammann et al., 2015)."

**Materials and methods**

• **Line 105 – From which four commercial companies?**

Response: This information has now been included, as follows:
"Seven biochars were obtained from the following feedstocks and produced at the
following temperatures: almond shell at 500 °C (AS500, produced by Karr Group Co.),
almond shell and 800 °C (AS800, Premier Mushroom and Community Power Co),
coconut shell at 650 °C (CS650, Cool Planet), softwood at 500 °C (SW500, Karr Group
Co.), softwood at 650 °C (SW650, Cool Planet), and softwood at 800 °C (SW800, Pacific
Biochar), and an additional softwood biochar produced at 500 °C and inoculated with a
proprietary microbial formula (SW500-I, Karr Group Co.)."

• **Line 107 – What is the inoculated microbial formula?**

Response: As the biochars are commercially available, many of the production details—including
the microbial formula—are proprietary and were not disclosed. However, now that we have
included the company names at the reviewers suggestion, other scientists can repeat
experiments with these biochars, working with the producers if desired.

• **Line 137 – Do not see the need to specify ongoing field trials if there is no connection with the**
**current paper.**

Response: The connection between these experiments and ongoing field trials is critical to the
novelty and importance of this study, as we are using the same soils and biochars to investigate
agronomically relevant responses at multiple scales. However, we agree that the connection
was not made clear enough, as description of the field trials is currently contained only in the
methods section 2.2. In the next iteration of this manuscript, we have included an additional
final paragraph in the discussion section, as detailed below:
"4.4 Implications for field conditions
It is difficult to extrapolate results from these laboratory-scale investigations to field-
scale, production agriculture, as real-world conditions will have additional variables in
climate, soil-water, and soil-plant dynamics. However, the results of this study suggest
these biochars may increase the residence time of water in sandy soils and increase
drainage in fine textured soils during irrigation or flooding events, or when soils are
otherwise saturated. Results may be particularly relevant for flooded agricultural
systems such as rice, where ammonium is the primary source of N and water retention
is a key parameter for success (Minami, 1995). Indeed, 95% of California rice production
occurs in the Sacramento Valley, where both the YSiL and HSL soils are common
(http://rice.ucanr.edu/About_California_Rice/). Data from these trials may help growers
in these regions and soil textures determine if biochar can increase water and nutrient
retention in their systems.
Recent meta-analyses have concluded that biochar substantially increased soil water
content at field capacity and permanent wilting point, in the field and lab, in coarse
textured soils only (Blanco-Canqui, 2017; Razzaghi et al., 2020). Despite these observed

| 170 | trends, benefits have also been observed in fine textured soils, including reduced crop |
| 171 | water stress, increased yield (Kerré et al., 2017; Nawaz et al., 2019), and reduced crop |
| 172 | loss during deficit irrigation (Madari et al., 2017). Other authors have reported little to |
| 173 | no effect, or transient effects, of biochar on soil water dynamics in both fine and coarse |
| 174 | textured soils (Jones et al., 2012; McDonald et al., 2019; Nelissen et al., 2015). However, |
| 175 | results from our experiments can only be conservatively extrapolated to dryland |
| 176 | agriculture or in soils that experience wet-dry cycles, as unsaturated hydraulic |
| 177 | conductivity was not measured. In order to determine how these biochars may behave |
| 178 | in unsaturated conditions, current three-year processing tomato field trials are currently |
| 179 | underway in these same soil textures, in which soil-water dynamics are being |
| 180 | measured." |

• **Line 156 – It makes more sense to present electrolyte concentrations on a mM or M basis, to**
**normalize it. Why is this test done with NaCl and the column tests with CaCl2?**

Response: We have changed the concentrations to mM at the reviewer's suggestion.
Monovalent electrolyte solutions are commonly used in sorption studies to avoid cation bridging
which would confound sorption results. However, Na is a known dispersing agent when added
to soils, and so $CaCl_2$ was used in column tests rather than NaCl to prevent dispersal and the
creation of preferential flow paths. We have added a statement about this in the materials and
methods section to make this reasoning transparent.

• **Line 165 – Which are the "multiple equations"?**

Response: We have edited this statement to say "Langmuir, Freundlich, and Langmuir-
Freundlich equations were tested to model the adsorption isotherms, with the Freundlich
equation (Eq. (2)) demonstrating the best fit based on $r^2$ values."

• **Line 175 – How were the columns packed?**

A citation for the packing has been added to increase replicability:

"Columns were prepared using the dry packing method according to Gibert et al.
(2014)."

• **Was the biochar homogeneously mixed with the soils?**

Response: Yes, soils and biochars were thoroughly and homogenously mixed through a
combination of stirring and shaking within a sealed container for a minimum of 120 seconds.
The following sentence has been added:
"Soils and biochars were thoroughly and homogenously mixed prior to being added to
tempe cells."

• **How was existence of preferential flow ruled out? Any tracer?**

We acknowledge that the use of a tracer would have been beneficial to our mechanistic
interpretation of results, and thank the reviewer for this suggestion. Even without a tracer,
however, there was no evidence of preferential flow in any of the five replicates for any of the
treatments. Error bars were very small for both nutrient concentrations across pore volumes in
the breakthrough curves, as well in the hydraulic conductivity measurements as measured by
data loggers.
• **What was the flow rate and the pore volume?**

Response: As stated on line 179, each column was gravity-fed a solution at a constant pressure
head of 34 cm. The "flow rate" is therefore the $K_{sat}$ itself, provided in the results section. Soil
porosity was provided in Table 3. Additionally, we have now edited the methods section to
provide core volume, as below in bold:

"To investigate the influence of biochar on saturated hydraulic conductivity ($K_{sat}$),
constant head column experiments were performed in five replicates using the 5 station
Chameleon Kit (Soilmoisture Equipment Corporation (SEC) 2816GX). SEC tempe cells**,**
**each with a volume of 136.4 cm$^3$** were packed with soils amended with 0 and 2% (w/w)
AS500, AS800, or SW500 biochars…"

• **Lines 175-184 – Why was ksat measured for 2 soils, whereas sorption for only 1?**

Response: We agree that the study would have benefited from leaching data from both the YSiL
and the HSL soils. However, column experiments were performed without the aid of
autosamplers or mechanization of any kind. As shown in Figure 3, the $K_{sat}$ for the unamended
HSL soil columns was 1.2 cm s$^{-1}$ . To manually collect 20 pore volumes of leachate, 15 hours of
active maintenance was required per treatment, for a total of 4 treatments. The YSiL $K_{sat}$ was
0.044 cm s$^{-1}$, or a 96% reduction in flow rate from the HSL. It was not feasible for someone to
stay in lab for the time required to collect 20 pore volumes at this speed. After attempting it for
two treatments, we decided to proceed with the HSL data, as it was logistically possible and
would provide more valuable information. To make this clear to readers, we will include the
following statement (new content in bold):
"Columns were also used to investigate the nutrient retention and leaching in HSL
amended with 0 and 2% biochar. Preliminary trials with the YSiL demonstrated that
leaching rates were very low (~0.044 cm$^{-1}$) creating logistical challenges for conducting
these experiments. Additionally, the impact of nitrate leaching is much more
pronounced in more coarsely textured soils and thus leaching experiments were
conducted only in HSL columns"

**Results**

• **Line 196 – increased pyrolysis temperature usually increases carbonization.**
We agree that increased temperature often results in higher carbon content; however, this is
dependent on the feedstock and production parameters, especially atmospheric oxygen
content. As these materials were obtained from commercial sources, we do not have specific information regarding oxygen levels. The higher ash content in the almond shell biochars was
expected, due to the high cation content of almond shell feedstocks (Aktas et al., 2015) which
are concentrated at higher temperatures. The impact of pyrolysis temperature on the ash
content of softwood biochars is less pronounced.
• **Line 268 – what do you mean by "main effect"? p values correspondence not clear.**

Response: We have edited the statement as follows:
"There was a significant effect of biochar ($p = 0.001$) and soil texture ($p < 0.001$), as well
as a significant interaction between biochar and soil texture ($p = 0.006$), on saturated
hydraulic conductivity."

• **Figure 4- It is very hard to discern the data and the decrease in leaching of NO3 from the**
**control to HSL+SW500. Please consider to reduce the y axis from 100 mg/L to something**
**smaller (4(a)) to make the graph better accessible (in regards to the data in the text) for the**
**readers.**

Response: We agree that the data is difficult to read in the one instance the reviewer indicates,
however, the y axis cannot be reduced, as the initial nitrate flush in pore volumes 0-10 neared
100 mg $L^{-1}$. As is, the figure shows the reader that a) HSL had very high but easily leached levels
of nitrate, and b) there was not much difference in nitrate leaching in soils with or without
biochar. This information is highly descriptive and necessary to the study. To provide a more
detailed snapshot of the data the reviewer is interested in, figure 5 was included so that exact
nitrate quantities could be obtained across pore volumes 15, 20, and 25. Together, these two
figures provide both an overview and a more fine-grained resolution of nutrient leaching in
these columns.

**Discussion: In general, the discussion is not sufficient, and needs better structuring, with more**
**references.**

Response: Our manuscript includes an extensive literature review with many references. We do not
believe we need *more* literature, but, as the reviewer stated, a *better structured* literature review
would be beneficial. Currently, we have included a lengthy discussion in the introduction. We did
not include these same references in the discussion so as to avoid repetition, but agree that this
context is important for our specific results. In the revised version, we have moved some of the
extensive discussion from the introduction into the discussion, and we have related all findings to
our results, as described below

• **Line 324-329 –This explanation is a bit underwhelming. A more mechanistic approach to this**
**would have been to also measure cations and anions in solution – if nitrate is bound to**
**positively charged components in the ash, one should see some anions being released. PZC**
**measurements would have been crucial in the experimental design, since a lot of the**
**reasoning is based on "electrostatic repulsion".**

Response: As described in line 68 of this document, we have now included PZC measurements
which substantiate the discussion of electrostatic repulsion and affinity.

• **Line 340-342 – A tracer study using "deuterated water" or something similar would have been**
**a more mechanistic way to explain the movement of water through biochar packed columns.**
Response: We acknowledge that the use of a tracer would have improved our mechanistic
interpretation of results, and thank the reviewer for this suggestion. In future work we will
consider this approach.

[revised manuscript text omitted]

---

## Author Response (AR2)

**Author's response**

We have made the suggested changes and thank the editor for their input and expertise. Specific details of our changes are outlined below.

**Topical editor comments:**

Most reviewer comments were addressed to a satisfactory degree, I have a few minor remarks before I can advise for final publication:

- Please explicitly address the concerns about preferential flow and the lack of a tracer in your experiments in the main manuscript.

  Response: We have added text to the end of the results section beginning on line 330: "Evaluation of data revealed no evidence of preferential flow in any replicate in any of the columns, which were carefully prepared according to the dry packing method in Gibert et al. (2014). This is demonstrated in the small error bars in figures of nutrient concentrations across pore volumes, as well as in hydraulic conductivity measurements taken by data loggers. To further monitor columns for preferential flow, the use of a conservative tracer could be considered in future experiments."

- The BET specific surface area is typically measured via $N_2$ physisorption, not $CO_2$. $CO_2$ physisorption (usually not analyzed via BET but other models) can be used as a complementary method for the characterization of micropores. For biochar which is expected to also contain mesopores and some macropores, $N_2$ is the agreed on standard method that can be complemented by $CO_2$ if needed. If you have indeed used $CO_2$ I suggest to also measure $N_2$ according to IBI as well as EBC standards. If the gas was actually $N_2$, please correct this accordingly.

  Response: The use of $N_2$ to measure surface area via the BET method is a standard approach which has long been used. However, significant drawbacks of this approach arise from the fact that the BET equation was developed to predict surface area of non-porous materials. $N_2$ cannot access pores < 0.5 nm, while $CO_2$ can. Additionally, $CO_2$ permits monolayer coverage and does not have the volume-filling effect which can arise with $N_2$. This information is presented in Sigmund et al., 2017. In an evaluation of 12 biochars analyzed for surface area using the BET equation and both $N_2$ and $CO_2$, results demonstrated numerous artifacts and unreliable results from $N_2$ isotherms (Maziarka et al., 2021). In that study, $CO_2$ showed greater efficacy for

detecting micropores with increasing highest temperature treatment (HTT) and, unlike $N_2$, did not show hysteresis in any sample. The use of $CO_2$ is routine for determining the surface areas of many carbon-based materials (e.g., activated carbon, biochar, carbon-based superconductors). In an analysis of carbon-based superconductors, $CO_2$ was shown to reveal the presence of pores in the ~1 nm range, probing pores much smaller than was possible with $N_2$. Additionally, $N_2$ isotherms demonstrated pore condensation and type H2 hysteresis (IUPAC classification) (Zhu et al., 2011). We believe that the use of $CO_2$ to measure surface is a sound approach, and that making comparisons between samples measured with the same methodology is appropriate.

The IBI and EBC standards were developed in 2015 as a means to standardize characterization for biochars. While these criteria have great value, we do not believe that the exclusive use of these methods is required for scientific research. We do not challenge the value of BET measurements with $N_2$, but do feel that using $CO_2$ is also commonplace and appropriate. The ideal scenario would be to complement $CO_2$ data with $N_2$ measurements, as neither method is perfect. While this is unfortunately not feasible for us to do at this time, we will consider this approach for future experiments.

**Editorial support team:**

- With the next revision, please re-name your "Figure S2" to "Figure S1".

  Response: We have made the suggested change.

**References**

Sigmund, G., Hüffer, T., Hofmann, T., & Kah, M. (2017). Biochar total surface area and total pore volume determined by N2 and CO2 physisorption are strongly influenced by degassing temperature. *Science of the Total Environment*, *580*. https://doi.org/10.1016/j.scitotenv.2016.12.023

Maziarka, P., Wurzer, C., Arauzo, P. J., Dieguez-Alonso, A., Mašek, O., & Ronsse, F. (2021). Do you BET on routine? The reliability of N2 physisorption for the quantitative assessment of biochar's surface area. *Chemical Engineering Journal*, *418*. https://doi.org/10.1016/j.cej.2021.129234

Zhu, Y., Murali, S., Stoller, M. D., Ganesh, K. J., Cai, W., Ferreira, P. J., Pirkle, A., Wallace, R. M., Cychosz, K. A., Thommes, M., Su, D., Stach, E. A., & Ruoff, R. S. (2011). Carbon-based supercapacitors produced by activation of graphene. *Science*, *332*(6037). https://doi.org/10.1126/science.1200770

---

## Author Response (AR3)

**Author's response**

We have addressed concerns regarding specific surface area measurements in the below text, as well as in the accompanying revised manuscript. We thank the editor for their input and expertise. Specific details of our changes are outlined below.

**Topical editor comments:**

Unfortunately, the authors have not addressed my concerns regarding specific surface area measurements in relation to the expected pore-size distribution in biochar, the applicability of the BET equation to CO2, as well as the comparability to standardized measurements according to international guidelines (IBI as well as EBC).Further, hysteresis in typical setups cannot be measured with CO2, as only adsorption and no desorption is measured when using CO2, which makes the hysteresis argument beside the point. The ISO norm mentioned is tailored to multilayer-adsorption gases such as nitrogen; CO2 is not mentioned in the cited guideline as far as I can tell. I advise the authors to re-consider their arguments for using CO2 and perhaps re-read the literature cited in their response (including the mentioned paper by this editor).

**Author's Response:**

We once again thank the editor for their input and expertise on the topic. We have reviewed our surface area data and communicated with the lab manager at Micromeritics, where we had the samples analyzed to clarify the methodology used. We were incorrect to report the surface area was calculated using the BET equation. The $CO_2$ isotherms at 273 K were used to determine micropore volume and micropore area ($SSA_{\mu p}$) using the non-local density functional theory (NLDFT) model specific for $CO_2$. The testing done by Micromeritics utilizes elements of ISO9277, such as sample preparation and aspects of the analysis approach; however, the BET method does not apply to $CO_2$ based measurements and was not used. These methodological nuances were not clear from the data initially returned to us by Micromeritics. We are grateful that the editor has specific expertise in this area, and thank them for their persistence with this aspect of our manuscript, as we can now more accurately report the methods used in our study. In our revised manuscript we specify that our values correspond to $SSA_{\mu p}$, instead of "SA", to make clear our data are specific to micropore surface area.

As the use of both $N_2$ and $CO_2$ to determine surface area each has distinct limitations, we agree with the editor that data from both measurements would provide complimentary information and a more quantitative understanding. We fully agree that the use of $CO_2$ has important limitations, specifically that it does not determine exterior surface area nor macropore surface area. Rather, this approach provides information of pores ranging, approximately, from 0.35 to 1.5 nm (or 3 nm if higher pressure is used). As the surface area

of many biochars is dominated by micropores, this method typically gives $SSA_{\mu p}$ values greater than SA measured via $N_2$ (e.g., Zeng et al., 2013; Maziarka et al., 2021). BET surface area with $N_2$ has a long history for surface area measurements and is indeed recommended by the International Biochar Initiative and the European Biochar Certificate for biochar characterization. However, this approach also has limitations, as it cannot access the microporosity of materials with pores < 0.5 um (e.g., Pignatello et al., 2006) (range for $N_2$: ~2 to 50 nm); it relies on assumptions that may not be true for small pores and materials with very high surface areas (Walton and Snurr, 2007); and pore flexing is limited at 77 K (too cold) and inhibits diffusion into micropores. These points are well stated by the editor in one of their publications (Sigmund et al., 2017).

It is commonly recommended that $N_2$ and $CO_2$ measurements both be conducted, to provide complimentary data regarding surface area of materials such as biochar. Although we do not have the BET surface area via $N_2$, we believe that reporting of the $SSA_{\mu p}$ provides meaningful information and is valuable in making comparisons between biochars and providing information for our study. Although not quantitative, we do have qualitative data regarding macropores as evidenced by X-ray microCT images. For future studies it would be best to have surface area measurements using both $N_2$ and $CO_2$ to provide complimentary analysis and "overcome" the limitations of each approach. We have edited our manuscript to reflect this information and hope that the editor will find our efforts satisfactory (new content in red):

**Methods**

, which are not probed via the $CO_2$ surface area approach The micropore specific surface area ($SSA_{\mu p}$) was determined from $CO_2$ adsorption isotherms at 273 K using the Non-Local Density Functional Theory (NLDFT) (Particle Testing Authority, Micromeritics TriStar II Plus 3.0). The micropore specific surface area ($SSA_{\mu p}$) was determined from $CO_2$ adsorption isotherms at 273 K using the Non-Local Density Functional Theory (NLDFT) (Particle Testing Authority, Micromeritics TriStar II Plus 3.0, NLDFT model mod11.df2). Prior to analysis, samples were degassed with $N_2$ at 393K for 16 h.

**Results**

**Line 221**: Softwood biochars produced at 500 and 800 °C had substantially higher SSAμp than almond shell biochars produced at the same temperatures. It should be noted, however, that SSAμp measured by CO2 adsorption frequently results in higher values than surface area measured by N2, as CO2 can access micropores unavailable to N2 (Maziarka et al., 2021; Zeng et al., 2013). While

results from each method tend to be well correlated and are considered to provide complementary information (Sigmund et al., 2017), neither should not be regarded as providing precise total surface area.

**Discussion**

**Line 438:** This is consistent with the increase in soil Ksat after addition of AS800 in YSiL, and the smaller effect of AS800 in HSL compared to AS500 and SW500 (discussed in section 4.3). Future investigation should include measurements of biochar surface area utilizing both $CO_2$ and $N_2$ adsorption. While $CO_2$ is commonly used to probe micropores in carbon-based materials (Maziarka et al., 2021; Sigmund et al., 2017; Zhu et al., 2011), IBI criteria recommends the use of $N_2$ for biochar analysis (International Biochar Initiative, 2015). Including $N_2$ measurements would aid in standardization across studies. Furthermore, the differences in results from each method may be descriptive of the relative pore size distribution between each biochar in this study. Differences in pore size distributions, as observed by X-ray microCT, have been demonstrated to have a varying effect on water retention and conductivity in previous studies (Devereux et al., 2013; Quin et al., 2014).

Other minor changes have been made to our manuscript, as indicated in the revised document with changes tracked.

**References:**

Pignatello, J.J., Kwon, S., Lu, Y., 2006. Effect of natural organic substances on the surface and adsorptive properties of environmental black carbon (char): attenuation of surface activity by humic and fulvic acids. *Environ. Sci. Technol*., **40**(24):7757-7763. [doi:10.1021/es061307m]

Ravikovitch, P.I., Neimark, A.V., 2001. Characterization of nanoporous materials from adsorption and desorption isotherms. *Coll. Surface A*, **187-188**:11-21. [doi:10.1016/S0927-7757(01)00614-8]

Gabriel Sigmund, Thorsten Hüffer, Thilo Hofmann, Melanie Kah. Biochar total surface area and total pore volume determined by N2 and CO2 physisorption are strongly influenced by degassing temperature. 2017. Science of The Total Environment, 580: 770-775, https://doi.org/0.1016/j.scitotenv.2016.12.023.

Walton, K.S., Snurr, R.Q., 2007. Applicability of the BET method for determining surface areas of microporous metal-organic frameworks. *J. Am. Chem. Soc*., **129**(27):8552-8556. [doi:10.1021/ja071174k]